# Sources of ultrafine particles at a rural Midland site in Switzerland

Lubna Dada[1*], Benjamin T. Brem[1*], Lidia-Marta Amarandi-Netedu[1], Martine Collaud Coen[2], Nikolaos Evangeliou[4], Christoph Hueglin[3], Nora Nowak[1], Robin Modini[1], Martin Steinbacher[3], Martin Gysel-Beer[1]

[1]PSI Center for Energy and Environmental Sciences, Villigen PSI, Switzerland

[2]Federal Office of Meteorology and Climatology, MeteoSwiss, Payerne, Switzerland

[3]Laboratory for Air Pollution and Environmental Technology, Swiss Federal Laboratories for Materials Science and Technology (Empa), Dübendorf, Switzerland

[4]Department of Atmospheric and Climate Research (ATMOS), Stiftelsen NILU, Kjeller, Norway

*Correspondence to: Lubna Dada (lubna.dada@psi.ch) and Benjamin T. Brem (benjamin.brem@psi.ch)

### Abstract

Ultrafine Particles (UFPs, i.e. atmospheric aerosol particles smaller than 100 nm in diameter) are known to be responsible for a series of adverse health effects as they can deposit in humans' bodies. So far, most field campaigns studying the sources of UFPs focused on urban environments. This study investigates the outdoor sources of UFPs at the atmospheric monitoring station in Payerne, which represents a typical rural location in Switzerland. We aim to quantify the primary and secondary fractions of UFPs based on specific measurements between July 2020 and July 2021 complementing a series of operational meteorological, trace gas and in-situ aerosol observations. To distinguish between primary and secondary contributions, we use a method that relies on measuring the fraction of non-volatile particles as a proxy for primary particles. We further compare our measurement results to previously established methods. We find that primary particles resulting from traffic and residential wood burning (direct emissions – mostly non-volatile BC-rich), contribute less than 40% to the total number of UFPs, mostly in the Aitken mode. On the other hand, we observe local new particle formation (NPF) events (observed from ~1 nm) evident from the increase of cluster ions (1.5 - 3 nm) and nucleation mode particles (2.5 - 25 nm) concentrations, especially in spring and summer. These events, mediated by sulfuric acid, contribute to increasing the UFPs number concentration, especially in the nucleation mode. Besides NPF, chemical processing of particles emitted from multiple sources (including traffic and residential wood burning) contribute substantially to the nucleation mode particle concentration. Under the present conditions investigated here, we find that secondary processes mediate the increase of UFP concentration to levels equivalent to those in urban locations, affecting both air quality and human health.

## 1 Introduction

Atmospheric aerosol particles are known for their adverse effects on human health and their impacts on Earth's climate. Among those, a class, namely ultrafine particles (UFPs – diameter < 100 nm), is critical, given the

particles' high number concentrations (and surface area) in the atmosphere and their capability to travel deep into the human body and to deposit onto sensitive body parts e.g. brain (Schraufnagel, 2020; Kwon et al., 2020). Therefore, scientists have dedicated a large fraction of research to advance the understanding of the chemical composition, sources and sinks of UFPs (Seinfeld and Pandis, 2016).

UFPs are found to be more abundant close to anthropogenic source locations. Studies conducted in Switzerland have demonstrated the prevalence of UFPs closer to major roads and airports (Meier et al., 2015; Eeftens et al., 2015; Rivas et al., 2020; Bukowiecki et al., 2002), highlighting the contribution of traffic-related emissions as a significant primary source to UFPs. Additionally, Switzerland's proximity to various industrial regions in Europe underscores the role of transboundary pollution in contributing to particulate matter levels (Zotter et al., 2014; FOEN, 2021). Composition-based source apportionment studies are typically done on mass basis for e.g. $PM_{2.5}$ and $PM_{10}$ (particulate matter with diameters less than 2.5 and 10 µm, respectively), yet some studies focused on UFP sources by number, utilizing the different patterns and shapes of the measured UFP size distributions (Grange et al., 2021; Trechera et al., 2023; Cai et al., 2024; Chen et al., 2022; Garcia-Marlès et al., 2024a; Garcia-Marlès et al., 2024b; Vörösmarty et al., 2024; Rivas et al., 2020; Kalkavouras et al., 2024).

Moreover, UFPs can form in the atmosphere from gaseous precursors via chemical processing. One example is new particle formation (NPF). This phenomenon is found to produce the dominant fraction of the total particle number concentration in several environments worldwide (Kerminen et al., 2018). The frequency and intensity of such events depend on various factors including meteorology, the availability of gaseous precursors and the levels of the preexisting particles acting as sinks (Nieminen et al., 2018). Previous studies on NPF have shown that sulfuric acid is the major precursor vapor contributing to the early steps of NPF within the boundary layer, especially in the presence of stabilizing bases such as ammonia and amines (Yan et al., 2021; Almeida et al., 2013; Kürten et al., 2016; Sihto et al., 2006; Dada et al., 2023). The latter are usually associated with agricultural and husbandry activities, although other sources such as traffic and industry have been reported in polluted megacities (Zhu et al., 2022). Given the abundance of agricultural activities in Switzerland, NPF is expected to be a major process and contributor to UFP. However, NPF field studies in Switzerland have been limited to short-term measurements in high-Alpine environments (at Jungfraujoch, 3580 m a.s.l.) (Bianchi et al., 2016; Manninen et al., 2010) leaving large gaps in our understanding of NPF in rural Switzerland and its contribution to the size-distribution of UFPs.

The contribution of the different sources to the size classes of UFPs (cluster mode: 1.5 – 3 nm, nucleation mode: 2.5 -25 nm, and Aitken mode: 25 – 100 nm) can vary depending on the type of the source, location, and time of year. For instance, combustion processes, including vehicle emissions, industrial activities, and biomass burning, typically produce UFPs rich in elemental carbon (soot) and organic compounds in the size range > 10 nm (Cai et al., 2020). Unfiltered diesel -gasoline direct injection- and gas turbine engines are known to emit a significant fraction of UFPs in the smaller size range, down to even few nm, due to their high combustion efficiency (Rönkkö et al., 2017; Maricq et al., 1999; Durdina et al., 2017) but also due to lube oil ash residues and nucleating volatile material (Kittelson et al., 2004). Other sources, natural or anthropogenic, e.g. road-dust resuspension, dust storms,

wind-blown pollen and plant debris, can mechanically produce coarse particles that have little to no contribution to UFPs. On the other hand, vapors associated with natural and anthropogenic activities, contribute to secondary aerosol formation and growth, substantially affecting the number size distribution of UFPs (Kulmala et al., 2004; Ma and Birmili, 2015; Sun et al., 2024).

In this study, we delve deeper into the outdoor sources of UFPs observed in Payerne, a typical rural location in Switzerland. We go beyond the traditional characterization of the site based on $PM_{2.5}$ by identifying and quantifying the sources of UFPs, be it primary or secondary. For doing so, we rely on comprehensive long-term particle (and ions) number distribution measurements at the location, starting from ~1 nm particle diameter. To distinguish between the size-segregated primary and secondary particles, we applied a measurement technique that uses a catalytic stripper. Together with available measurements of meteorological parameters and trace gases, we could characterize (secondary) NPF events at Payerne and quantify their frequency and intensity, as well as the role of agriculture in providing the vapors responsible for their occurrence, and the contribution of such events to the UFPs at the location. Besides NPF, the secondary fraction of particles resulting from the atmospheric processing of available particles (be it primary or secondary) was quantified. The identification and quantification of primary and secondary sources of UFPs, in such a typical rural location in Switzerland, are pivotal steps allowing for targeted interventions to control UFP emissions, reduce pollution and thus exposure risks.

## 2 Materials and Methods

### 2.1 Measurement location

The measurements were conducted at MeteoSwiss' aerological station Payerne (46.812 N, 6.942 E, 491 m a.s.l.) in canton of Vaud, Switzerland. The station hosts long-term observations for a variety of programmes like SwissMetNet, ACTRIS (The Aerosol, Clouds and Trace Gases Research Infrastructure), NABEL (National Air Pollution Monitoring Network), BSRN (Baseline Surface Radiation Network), GAW (Global Atmosphere Watch Program) and GRUAN (GCOS Reference Upper-Air Network). The station is about 1 km south-east of the rural town of Payerne and roughly 500 m north of a major regional road and about 4 km south of the national highway (A-1) and Payerne military airport (Fig. S1A). The city of Payerne has a total area of 24.19 $km^2$ and a population of roughly 10,500 people, in 2023. The Tropospheric Ozone Assessment Report Activity categorized the Payerne station as a rural site, based on satellite-retrieved $NO_2$ column concentrations, the surrounding population density, and the intensity of nighttime lights (Schultz et al., 2017). The Payerne site is also part of the European Monitoring and Evaluation Programme (EMEP) (Tørseth et al., 2023). EMEP's siting criteria require large spatial representativeness meaning that monitoring sites are only little influenced by local processes such as emissions, sinks, or topographic features, etc. (EMEP/CCC, 2001). Consequently, lower aerosol mass concentration compared to the other urban and suburban stations of the NABEL network is observed - an average hourly $PM_{2.5}$ concentration of 9.2 ± 8.0 µg/m$^3$ during our measurement period. As for the site itself, shown from the land use map (Fig. S1B), croplands and grasslands surround the measurement station resulting in contributions from

agricultural practices. The station is located slightly uphill and elevated (~40 m) above the adjacent emission sources (city, roads and airport).

## 2.2    Measurement instrumentation and details

A summary of all instruments used for this study are shown in Table S1. The investigated period covers the time between July 2020 and July 2021 when the routinely performed monitoring activities were complemented by additional aerosol number size distribution and NPF analyses. All data shown below are reported in Central European Time (CET; i.e. UTC+1).

### 2.2.1    Meteorology and Boundary Layer Height

In-situ global radiation, temperature, precipitation, wind speed and wind direction data were obtained from the SwissMetNet (https://opendata.swiss/de/dataset/automatische-meteorologische-bodenmessstationen, last access 26.09.2024), the automatic measurement network of MeteoSwiss. Temperature and wind speed profiles originates from the EMER-Met (Emergency Response Meteorology) network. The mixing layer height (MLH) was estimated from the temperature profile measured by the MicroWave Radiometer (MWR, HATPRO-G2 produced by RPG Radiometer Physics Gmbh) and the wind speed profiles measured by the Radar Wind Profiler (WP, PCL1300 produced by Degreane) by the bulk Richardson method that takes potential energy and wind shear into account (Collaud Coen et al., 2014). During nighttime, when the MLH is lower than the first level of measurement of the MWR, the data are unavailable resulting in a slight overestimation of the average MLH.

### 2.2.2    Trace gases

A suite of trace gases such as ozone ($O_3$), nitrogen dioxide ($NO_2$), sulfur dioxide ($SO_2$) and ammonia ($NH_3$) were measured at Payerne (Bundesamt für Umwelt, 2023). For the period of interest here, surface $O_3$ was monitored by UV absorption photometry (Thermo Scientific 49i), $NO_2$ was measured by chemiluminescence (Thermo Scientific 42i TL), $SO_2$ was measured by UV fluorescence (Thermo Scientific 43i TLE), and $NH_3$ was detected by Cavity Ringdown Spectroscopy (Picarro Inc., G2103). Details about the measurement procedures for the individual species can be found in the NABEL's technical report (EMPA & BAFU, 2023). The terminology 'concentration' used throughout the manuscript refers to the mixing ratio of the trace gases, in ppb, unless specified otherwise.

### 2.2.3    Particle number size distribution measurements

The particle number size distribution between 2.5 and 470 nm was measured using a Neutral Cluster and Air Ion Spectrometer (NAIS) and Scanning Mobility Particle Sizer (SMPS) placed in parallel. The NAIS measured the particle number size distributions between 2.5 and 42 nm, whereas the SMPS (TSI3034) measured the size range between 10 and 487 nm, in 3-minute time resolution. A condensation particle counter was also deployed, the CPC (CPC 3022) with a lower cut-of diameter ($D_{50\%}$) of 7 nm, with 1 second time resolution. In addition, the number size distribution of naturally charged positive and negative ions in the size range of 0.8 – 42 nm were measured using the NAIS.

The SMPS data inversion software accounted for multiple charged particles. Furthermore, a size dependent correction factor for the CPC counting efficiency and particle losses within the instrument and inlet lines was applied to the data (Liscinsky and Hollick, 2010; Yook and Pui, 2005). The NAIS is expected to overestimate the particle concentration by up to a factor of 10, with charging being the major source of uncertainty (Kangasluoma et al., 2020; Gagne et al., 2011). Here, the ratio of SMPS to NAIS in the overlapping size range 20 – 30 nm (highest detection efficiency size range for both instruments) was used to derive a correction factor for the NAIS. Based on the 50% cumulative distribution function of the ratio, the NAIS was found to overestimate the concentration by a factor 3.5 (25[th] and 75[th] percentiles were 2.5 and 5.8, respectively). For consistency, a constant factor of 3.5 was applied to the entire NAIS data set before combining the data with the SMPS. An example, during an NPF event and another during a nonevent day, are shown in Fig. S2, respectively. The overlapping size-cut was defined to be 25 nm until May 31[st] and 12 nm after that due to unreliable data from the NAIS for diameters >12 nm resulting from a dirty malfunctioning electrometer. NAIS data were available between 09 October 2020 and 31 July 2021. Please refer to table S1 for data availability.

We define four classes of particles based on the combined size distribution measurements. Ultrafine particles are defined as all particles with diameters less than 100 nm, Aitken mode particles have diameters between 25 and 100 nm, nucleation mode particles have diameters between 2.5 and 25 nm, and cluster ions have diameters between 1.5 and 3 nm.

### 2.2.4    Particle number size distribution of non-volatile particles

An additional SMPS (TSI3938) was placed behind a catalytic stripper to measure the non-volatile particles in the size range 6 – 110 nm, in 1 minute time resolution. The catalytic stripper (CS015, Catalytic Instruments GmbH) has an operating gas temperature of $350^{\circ}$C which evaporates the volatile particles allowing the fraction of non-volatile particles to be measured by the SMPS. The design of the instrument used is based on a design widely used in emissions measurements in the automotive industry, where strict requirements exist for volatile removal to report solid particle numbers. This instrument can remove more than 99.9% of 30 nm tetracontane particles at inlet mass concentrations up to 1 mg/m³ (Andersson et al., 2007). While such extreme exhaust conditions - characterized by very low volatile material and high particle concentrations - are not observed in Payerne, the volatile removal efficiency is expected to be even higher for the data we report. The catalytic stripper was combined with an automated switching system that alternated, automatically, between total and non-volatile particles measurements every 5 minutes. Non-volatile size-distribution data were available between 19 December 2020 and 03 March 2021. Please refer to table S1 for data availability.

### 2.2.5    Equivalent black carbon

Equivalent Black Carbon concentrations were determined from an aethalometer (Model AE31; Magee Scientific Inc.) measuring at seven wavelengths (370, 470, 520, 590, 660, 880, and 950 nm), with a time resolution of 5 minutes. In this study, we use the 880 nm channel, hereafter, referred to as Black Carbon (BC). Data processing

followed GAW/ACTRIS recommendations and included corrections for filter loading and multi-scattering effects (Zanatta et al., 2016).

## 3 Data analysis

### 3.1 Brightness parameter

The brightness parameter ($B$) is a parameter used to *estimate* the degree of cloudiness at a certain location. It is defined by the fraction of the total solar radiation reaching the measurement site where the only blockage considered is existing clouds (Dada et al., 2017; Sánchez et al., 2012). Hence, the ratio of the measured global radiation to the theoretical maximum solar radiation at the top of the atmosphere are used to calculate $B$:

$$B = \frac{Global\ Radiation}{Theoretical\ Maximum} \tag{1}$$

The larger the $B$ value, the less clouds are in the sky and more radiation arrives to ground level. A complete cloud cover is defined as $B < 0.3$ and clear sky is defined as $B > 0.7$.

### 3.2 BC concentration as a tracer for primary particles

For determining the relative contributions of primary and secondary particles to the size segregated particle concentrations over the full year of measurements, we use the empirical method suggested by Rodríguez and Cuevas (2007) and modified by Kulmala et al. (2016) that uses BC mass concentration as a tracer for primary particles. Using this method, the concentration of primary particles, $N_{primary}$, is estimated from measured BC concentrations using the following equation:

$$N_{primary} = S_1 \times BC \tag{2}$$

Here, $S_1$ is the semi-empirical scaling factor derived from the particle concentration to BC ratio. $S_1$ is determined as the minimal $N$-to-BC ratio, or a low percentile (e.g. 0.1%, 1% or 5%) of all observed N-to-BC ratios. The concentration of secondary particles, $N_{secondary}$, is obtained from the difference of total particle number concentration in the selected mode ($N$) and the primary fraction ($N_{primary}$):

$$N_{secondary} = N - N_{primary} \tag{3}$$

The fractions of primary and secondary ultrafine and accumulation mode particles derived from this method were compared to the corresponding fractions determined by the non-volatile particle measurement method described in section 2.2.4 over the 3-month period when the catalytic stripper (section 2.2.4) was deployed.

### 3.3 New particle formation events classification

Days in Payerne were classified into NPF event or nonevent days depending on the evolution of their particle number size distributions. Here, we followed the traditional method introduced by Dal Maso et al. (2005), and modified by Dada et al. (2018) to distinguish between local and transported events based on the size distributions

below 3 nm. Here, we combined both aforementioned methods yet tailored them to fit our measurement location better, which is subject to traffic emissions, as the previous two methods were developed for the boreal forest environment. Our classification scheme is shown in Fig. S3. In brief, NPF events are recognized by the appearance of particles in the nucleation mode exhibiting signs of growth. Five classes were defined in our study. *Local events* (1) are those observed to start in the sub-3 nm range (i.e., hourly-averaged concentration of $1.5 - 3$ nm ions $> 20$ $cm^{-3}$), while *transported events* (2) are those observed starting at larger sizes (i.e., hourly-averaged concentration of $7 - 25$ nm particles $> 3000$ $cm^{-3}$) indicating their transportation to the measurement location (horizontally or vertically). Alternatively, some events are observed to start at small sizes (sub-3 nm) but fail to grow past a couple of nm. These events are referred to as *bumps* (3). Days during which none of the aforementioned phenomena were observed are considered *nonevent days* (4). Days which are affected by rain (rain rate $> 0$ mm) or high traffic influence (determined as $NO_2$ concentrations $> 3$ ppb) are classified as *undefined days* (5), as an NPF event cannot be visually inferred. We note that, the traditional regional events known as 'banana-shaped' events could be either (a) local + transported, i.e. we observe the evolution of the particles starting from the sub-3 nm diameters and growing into the shape of the banana where they merge with regional NPF events, or (b) only transported to our site where they are observed as a regional event but do not have a tail extending to the sub 3 nm region. In the case of Payerne, we did not observe any events that did not extend to the sub 3 nm size range, and hence all regional events are a combination of local nucleation and transported events.

## 3.4    Condensation and Coagulation sinks

Condensation sink, CS, is the rate at which gaseous precursors are lost to pre-existing particles. Here, the CS for sulfuric acid was calculated using the SMPS data following the method described in Kulmala et al. (2001) and Kulmala et al. (2012). The coagulation sink (CoagS) describes the rate at which freshly formed particles are lost to pre-existing particles. In this study, the CoagS was calculated using the combined size distribution (see section 2.2.3) following the method described in Kulmala et al. (2001) and was used an input for the calculation of particle formation rates, section 3.6. The equations for calculating CS and CoagS are shown in the SI.

## 3.5    Particle growth rates

Particle apparent growth rates (GR) were calculated using the 50% appearance time method from the positive ions number size distribution data measured by the NAIS (Dada et al., 2020a). Positive ions were chosen for the GR calculation as those have been found more important than the negative ion when it comes to ion induced nucleation from biogenic precursors (Baalbaki et al., 2021; Bianchi et al., 2021). In addition, in a location such as Payerne, where the dominant nucleation mechanism is neutral $H_2SO_4$-amine clustering (shown later in section 4.4.3), the transition from charged to neutral clusters is very short resulting in little to no difference between the growth rates retrieved from charged and those retrieved from total (charged + neutral) particles (Huang et al., 2022; Gonzalez Carracedo et al., 2022).The method relies on the determining the times when the concentration in each size bin reaches 50% of the maximum concentration. In this study, the GR for the size classes $1.5 - 3$ nm, $3 - 7$ nm and $7 - 15$ nm were determined. The growth rates were also used in the formation rates' calculation in the next section.

## 3.6 Particle formation rates

Particle formation rates at 2.5 nm ($J_{2.5}$) were calculated using the balance equation described in Kulmala et al. (2012) where the change in concentration of particles within a certain size bin (here 2.5 – 7 nm) depends on the particle sources (NPF) and the available sinks; here coagulation sink (CoagS$_{Dp}$) and growth out of the size bin.

$$ J_{Dp} = \frac{dN_{Dp}}{dt} + CoagS_{Dp} \cdot N_{Dp} + \frac{GR}{\Delta Dp} \cdot N_{Dp} \tag{4} $$

$Dp$ represents the lower diameter of the bin (here 2.5 nm), $N_{Dp}$ is the particle number concentration inside the size bin (2.5 - 7 nm), and GR is the growth rate of particles out of the bin (GR$_7$). $\Delta Dp$ is the difference between the upper and lower ends of the size bin of interest (here $\Delta Dp$ =4.5 nm). However, during the events for which a GR could not be calculated, given the change of air mass or the interruption of the growth, or during nonevent days, we used a median growth rate of all the events in the same month to estimate the formation rate. Such an estimation is valid given the similarity in particle GR regardless of the occurrence or intensity of NPF, see also Kulmala et al. (2022).

## 3.7 Extrapolation of particle formation rates

The particle formation rates at $J_{1.5}$ were extrapolated from $J_{2.5}$ following the analytical formula derived by Kerminen and Kulmala (2002):

$$ J_{dp_1} = J_{dp_2} \times \exp\left( -\gamma \, \frac{CS'}{GR_{dp_2 - dp_1}} \left( \frac{1}{dp_2} - \frac{1}{dp_1} \right) \right) \tag{5} $$

where $\gamma$ is a coefficient with an approximate value of 0.23 m$^3$ nm$^2$ s$^{-1}$ derived from the ratio between GR and CS′ (Eq. 11 and 13 in Kerminen and Kulmala (2002)). Here $dp_1 = 1.5$ nm, $dp_2 = 2.5$ and GR between 1.5 and 3 nm is calculated as GR$_{2.5-7}$ divided by 3 (Kulmala et al., 2013). In addition, CS'=4$\pi$D/CS; where CS is the condensation sink, described in section 3.4, and $D$ is the sulfuric acid diffusion coefficient.

We find that the exponent term in our case, where CS$_{average}$ = 0.005 and GR$_{2.5-7 \, nm(average)}$ = 3.4 nm/h, is 1.0002, and hence expect that $J_{2.5} \sim J_{1.5}$. In Fig. S4, we show a histogram of the ratio of $J_{1.5}$ to $J_{2.5}$ in 5 minutes time steps and find that the ratio falls between 1 and 1.04, concentrated at 1. Therefore, for the rest of the analysis, $J_{1.5}$ is approximated as measured $J_{2.5}$.

## 3.8 Sulfuric acid proxy (H$_2$SO$_{4,proxy}$)

The sulfuric acid proxy was calculated based on the sources and sinks of sulfuric acid in a rural environment setting as discussed in Dada et al. (2020b). The main source of H$_2$SO$_4$ in the gas phase is the oxidation of SO$_2$ by the hydroxyl radical (OH) while its sinks are mainly the loss of H$_2$SO$_4$ onto pre-existing particles, CS, as well as its loss to cluster/particle formation (Dada et al., 2020b). Since OH concentrations were not measured due to

instrument limitations, global radiation was used as an OH proxy (Dada et al., 2020b; Petäjä et al., 2009). The equation is as follows:

$$[H_2SO_4] = -\frac{CS}{2k_3} + \sqrt{\left(\frac{CS}{2k_3}\right)^2 + \frac{[SO_2]}{k_3}(k_1 \times \boldsymbol{Global\ Radiation})} \tag{6}$$

where $k_1$ and $k_3$ are coefficients specific for rural locations, represent the $H_2SO_4$ production from $SO_2$ in the presence of radiation and loss of $H_2SO_4$ to clustering, and are 0.92 x $10^{-8}$ $m^2W^{-1}s^{-1}$ and 2.21 x $10^{-9}$ $cm^3s^{-1}$, respectively (Dada et al., 2020b).

## 3.9    Contribution of NPF to UFP

The contribution of NPF events to UFP concentration ($\bar{N}_{NPF}$) could be calculated as suggested by Sun et al. (2024):

$$\bar{N}_{NPF} = \frac{\bar{N}_{NPF-HR} \times n_{NPF-HR} + \bar{N}_{NPF-LR} \times n_{NPF-LR}}{n_{NPF-HR} + n_{NPF-LR} + n_{NON-HR} + n_{NON-LR}} \tag{7}$$

where $\bar{N}_{NPF-HR}(\bar{N}_{NPF-LR})$ are the integrated diurnal of the difference between UFP concentration between events and nonevents under high radiation (low radiation). $n_{NPF-HR}$ ($n_{NPF-LR}$) and $n_{NON-HR}$ ($n_{NON-LR}$) are the number of high-radiation (low-radiation) days with and without NPF events, respectively. We note that similar to Sun et al. (2024), days are separated into high-solar-radiation and low-solar-radiation days by a threshold of daily average global radiation of 100 W m$^{-2}$.

## 4    Results

### 4.1    Characterization of Payerne site

#### 4.1.1    Meteorology

Payerne, a typical rural location in Swiss midlands, experiences a temperate climate characterized by distinct seasonal variations in terms of temperature and precipitation (Fig. S5 – S6). Winter, between December and February, is relatively dry compared to spring, between March and May, which witnesses increased precipitation and the emergence of blooming vegetation. Summer, spanning from June to August, is occasionally interrupted by short bursts of heavy rain from thunderstorms occurring mostly in the afternoon. Autumn, covering September to November experiences increased precipitation, as temperatures cool. Solar radiation and cloudiness levels follow the seasonal pattern, with higher levels in summer and lower levels in winter and autumn (Fig. S5 – S6).

#### 4.1.2    Variability of trace gases

The diurnal cycles (Fig. 1) and the time series (Fig. S7) of the trace gases with dominant sources from fossil fuel combustion, traffic, residential wood burning and industrial activities ($NO_2$, $SO_2$) reflect the rural conditions at Payerne. Concentrations are lower compared to ones observed at urban locations within the Swiss national

network (Bundesamt für Umwelt, 2023). However, short-term variability in the time series and the diurnal patterns indicate some influence from local to regional processes.

Ozone concentrations are influenced by various factors, such as sunlight, temperature, and concentrations of other pollutants such as volatile organic compounds and $NO_2$, with generally higher levels observed in spring and summer. For the gaseous pollutants $NO_2$ and $SO_2$, it is common for rural areas to have lower concentrations compared to urban environments (EEA, 2023), yet, their concentration can be influenced by various factors, including agricultural practices, traffic, local industrial activities, and long-range transport (Jion et al., 2023; EEA, 2023). Traffic emissions are a major source of $NO_2$ indicated by the sharp increase of $NO_2$ during the morning and evening rush hour (Fig. 1, Fig. S7), although other sources such as residential wood burning and use of fertilizers in agriculture could affect the concentration (Jion et al., 2023). $NO_2$ is observed in higher concentrations on workdays compared to weekends in line with the higher traffic volumes on workdays (Fig. S8B), further confirming the major contribution of traffic emissions to $NO_2$ concentration in Payerne. The variation between the different seasons is attributed to a shallower boundary layer in winter leading to accumulation of $NO_2$, compared to better dispersion and vertical mixing during summer (Fig. S9).

Compared to the nitrogen oxides, $SO_2$ concentrations are influenced by fossil fuel combustion but also long-range transport (Vestreng et al., 2007). Given the proximity of the Payerne airport to our measurement location, we inspected the direct contribution of airport emissions on $SO_2$ concentrations. We note that Payerne Airport handles an average of 754 civil flights per year, significantly fewer than Zurich Airport, which accommodates around 22,300 flights annually (averages for 2019-2021, Business Aviation Study Switzerland 2022). However, Payerne is the major military airport of Switzerland and has a lot military flight operations. Specifically, we: 1) compared the $SO_2$ concentrations during workdays and weekends (Fig. S8A), 2) inspected the $SO_2$ concentration when the wind direction arriving to our measurement station is from the airport (Fig. S8C), 3) studied the $SO_2$ concentrations during periods of airport holidays (Fig. S8D), and 4) compared the concentrations of $SO_2$ in Payerne to those at Rigi-Seebodenalp which is approximately 185 km away from Payerne and at an altitude of 1031 m a.s.l. (Fig. S8E). Our results show that while we cannot completely rule out the contribution of the airport activities on $SO_2$ concentrations observed at Payerne station, the evidence indicates that there is no substantial enhancement of $SO_2$ when airport activities are taking place. For instance, we observe similar levels of $SO_2$ on workdays and weekends (Fig. S8A), compared to the lower median $NO_x$ concentrations on weekends compared to workdays (Fig. S8B), which are associated with a decrease in traffic intensity. However, we do observe a decrease in the extreme concentrations of $SO_2$ on weekends compared to workdays (Fig. S8A). Moreover, although wind arriving from the airport direction (NW: 300 – 340˚) does show elevated $SO_2$ concentrations, the higher concentrations are not exclusively related to this particular wind direction (Fig. S8C), as Payerne village is also located north-northwest to our measurement site (Fig. S1). Similar to our observation during weekends, $SO_2$ concentrations do not drop significantly during airport shutdown periods (Fig. S8D), particularly during 2020. Overall, these observations point towards other sources of $SO_2$ emissions, mostly related to energy use and supply (EEA, 2023). The latter conclusion is further verified by direct comparison between $SO_2$ concentrations in Payerne and Rigi-Seebodenalp. Similar $SO_2$ levels are observed at the two sites throughout the year except during winter, when higher

concentrations are observed in Payerne due to a shallower mixing layer height (Fig. S9) together with increased local SO$_2$ emissions from heating sources (EEA, 2023).

On the other hand, ammonia (NH$_3$) is expected to show higher concentrations in rural environments subject to agricultural practices, such as manure and fertilizers applications (Reche et al., 2022; Grange et al., 2023). In Payerne, the highest NH$_3$ concentrations are observed in spring, with a sharp peak in March when farmers prepare the first fertilization after winter and snow melt (Fig. 1, Fig. S7). In addition, warmer temperatures and longer days promote the release of ammonia (Fig. S10) (Pedersen et al., 2021). The distinct diurnal pattern of NH$_3$ concentrations during spring, i.e. earlier morning peak and increase during nighttime, could be attributed to farming activities such as fertilization and grazing, which peak at dawn and dusk. In autumn, as harvesting is completed, we observe a decrease in ammonia emissions, although post-harvest practices and the decomposition of crop residues can still contribute to the overall NH$_3$ concentration. Winter generally witnesses reduced agricultural activity, leading to lower ammonia emissions. The colder temperatures limit volatilization from soil and other surfaces however other sources such as traffic and fossil fuel combustion which emit NH$_3$ as a byproduct might become more important (Reche et al., 2022).

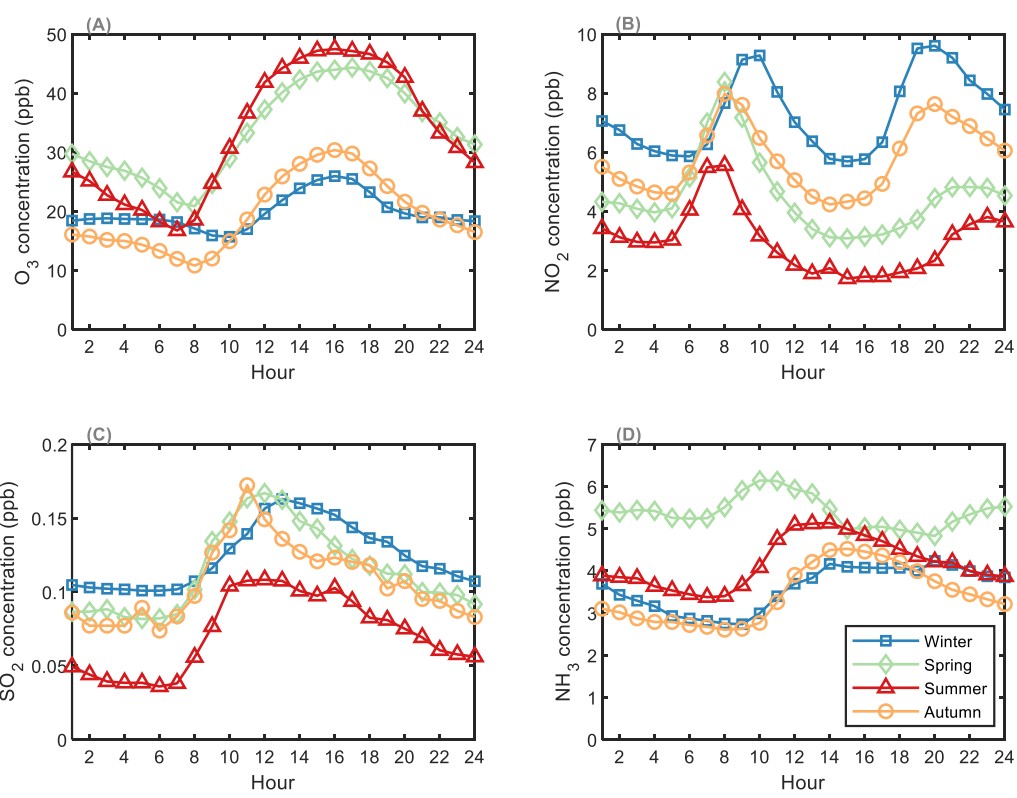

**Figure 1 Seasonal diurnal averages of trace gases in Payerne. See Fig. S6 for the full year time series.**

## 4.2 Ultrafine particles at Payerne

UFP particles in Payerne have multiple sources, some of which are primary (e.g. traffic) or secondary (e.g. NPF). Such sources are expected to have a strong diurnal and day to day variability as we observe a strong variation in the number concentration within each month sometimes exceeding two orders of magnitude (Fig. S11). Regardless, throughout the year, we observe a constant background of ~4000 cm$^{-3}$ UFP (Fig. 2A) along with distinct diurnal patterns on top of this background suggesting considerable contribution from local UFP sources. Our measured UFP concentrations resemble those measured in rural and suburban locations around Europe (Trechera et al., 2023). The Aitken mode contributes around half of the total UFP background and shows very little seasonal variability (Fig. 2C). In contrast, the nucleation mode, which also contributes about half to the UFP background, exhibits a distinct increase during daytime, making it the dominant contributor to total UFP and driving the variability in UFP concentration, especially in spring (62.4%) (Fig. 2B). This observation indicates an important role of nearby emissions and secondary aerosol formation relative to long range transport. During all seasons, the morning and evening peaks are concurrent with rush hours and could be attributed to traffic emissions, visible also in both the nucleation and to a lesser extent in the Aitken modes, in line with previous studies showing the contribution of traffic emissions to these size classes (Zhou et al., 2020; Rönkkö et al., 2017). An additional midday peak in the UFP concentration, driven by nucleation mode particles, is observed in spring and summer indicative of NPF events taking place in warmer seasons. This conclusion is further verified by the increase of cluster ions in these months (Fig. 2D), which are indicative for local NPF, usually more abundant in spring (Dada et al., 2023; Dada et al., 2017) following the onset of vegetation, warmer temperatures and increased solar radiation (Fig. S6). A detailed analysis of NPF events is presented in section 4.4. Although UFP number concentration data for autumn are incomplete (no data is available for the months of August and September), we expect a midday peak in the UFP concentration as observed in the Aitken mode particles – in August (Fig. S12). Given the nature of the location in Payerne (see Fig. S1), we expect the harvesting season to affect the concentration not only of the coarse particles but also of the ultrafine ones. In addition, earlier studies have shown a second wave of NPF events occurring during autumn when temperatures decrease allowing for better condensation of gases into the particle phase (Dada et al., 2017). In the next sections we quantify the contribution of the primary and secondary sources to the total UFP concentrations.

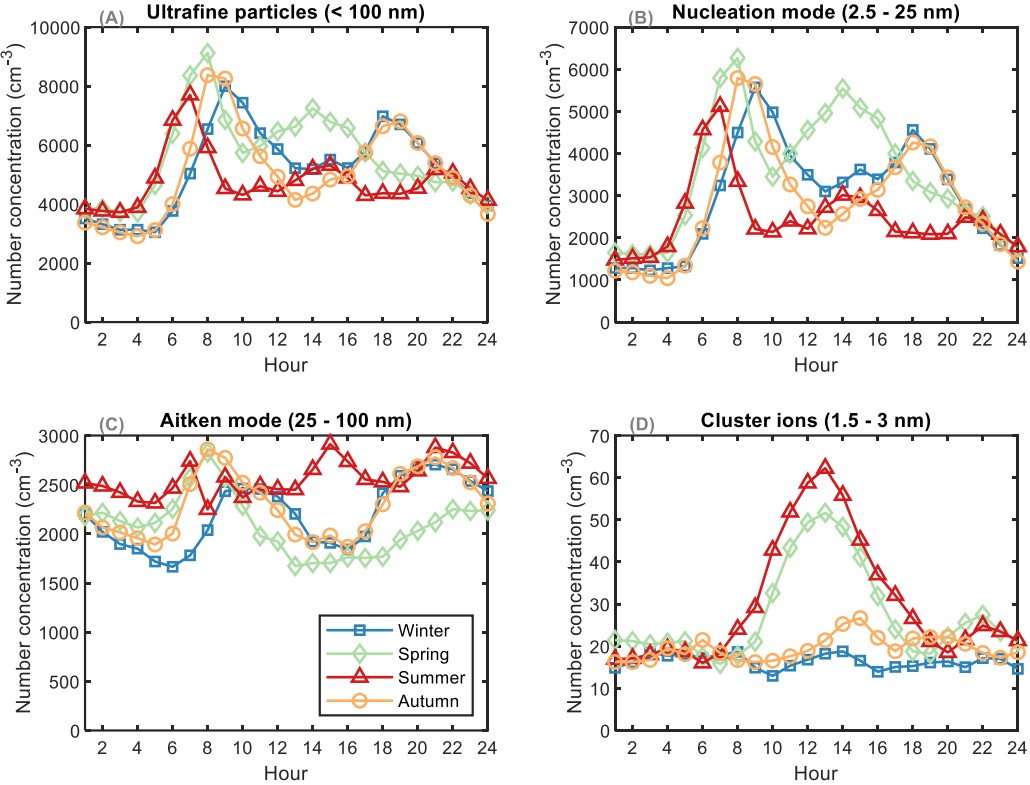

**Figure 2** Seasonal diurnal average concentrations of (A) ultrafine, (B) nucleation mode and (C) Aitken mode particles and (D) cluster ions (positive) in Payerne. No data from August and September is available for the ultrafine and nucleation mode particles or cluster ions. For the Aitken mode particles, July includes 2020 and 2021 concentrations. Data included in ultrafine, nucleation mode and cluster ion mode concentrations are in 5 min time steps (NAIS time stamp), while the Aitken mode concentrations are in 3 min time steps (SMPS time stamp). See monthly diurnal variations in Fig. S11.

## 4.3    Primary UFP at Payerne

### 4.3.1    Non-volatile particles

The exact differentiation between primary and secondary particles is difficult at a receptor site because it depends on a lot of factors such as residence time, precursors etc. Nevertheless, the size-segregated number concentration measurement with the catalytic stripper allows for a rough estimation of primary particles, since these non-volatile particles are expected to consist mainly of refractory black carbon from incomplete combustion processes (Wang et al., 2018), mostly from traffic and wood burning emissions. The secondary particles are in this study approximated as the difference between total and non-volatile particles. However, this approximation results in an overestimation of the secondary particles, since not all volatile particles necessarily are secondary in nature as for example traffic also emits primary volatile and semi-volatile UFPs (Baltensperger et al., 2002; Saarikoski et al., 2023). In Fig. 3, we show the particle number size distribution data retrieved from the total measurement, the non-volatile measurement, and their difference (secondary particles) for an example day with a strong NPF event (17.02.2021). Based on our observation, the morning and evening traffic rush hours considerably increase the concentration of non-volatile particles consistent with our expectations that traffic emits black carbon particles.

Photo-chemically driven NPF starting before noon cause a distinct increase of total UFP concentration (Fig. 3A), which is entirely dominated by volatile particles as expected (see difference plot in Fig. 3C). The size distributions of the non-volatile and secondary particles show distinct behavior in which the secondary particles contribute to smaller particles' concentration in the nucleation mode and reach higher diameters because of condensational growth. Secondary processes involving condensation facilitate the growth of particles until they reach climate relevant sizes, where they can contribute to regional scale phenomena e.g., regional NPF and cloud condensation nuclei (Zhou et al., 2021; Dai et al., 2017).

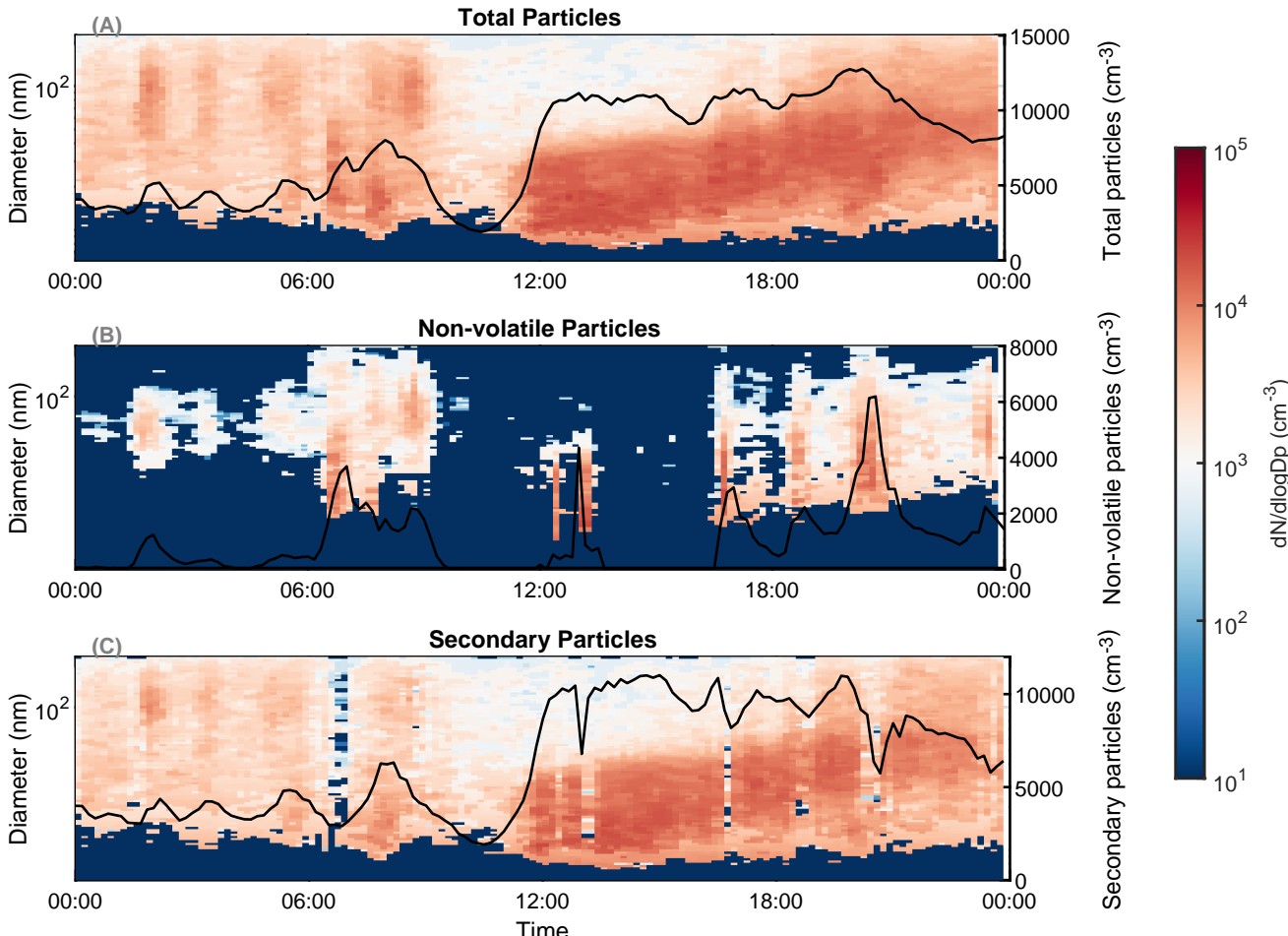

**Figure 3 Particle number size distribution from (A) the total measurement, (B) the non-volatile measurement and (C) their difference (secondary particles) during a strong NPF event on 17.02.2021. The black solid lines relevant for the right axis show integrated number concentrations.**

Using the ~3 months of available measurements (19.12.2020 – 3.3.2021), we quantified the contribution of primary particles to the total UFPs and to each of the modes (Fig. 4, S13). We find a dominance of the secondary

particles on most days, except for few cases (discussed in detail in section 4.3.3). However, this method is subject to limitations including:

1) Uncertainty on primary particles' number: quantifying primary particles through non-volatile particle measurements produces in general a systematic low bias in estimated primary particle number as primary volatile particles are misclassified as secondary particles. However, with a smaller likelihood, under certain circumstances, the uncertainty of the primary fraction could be further compounded the opposite way by the presence of very low volatile secondary particles that survive temperatures greater than 350°C (Wu et al., 2017; Kalberer et al., 2004).

2) Uncertainty on the size-distribution: Denuding volatile particulate matter from internally mixed particles with a non-volatile core reduces their diameter. This affects the size distribution of secondary particle calculated as the difference from total and non-volatile particle size distributions. Specifically, small secondary particles may be underestimated, whereas larger secondary particles may be overestimated if a substantial fraction of internally mixed primary non-volatile particles with a secondary coating is present.

3) Measurements only during a short period in winter might not be representative of the full year.

Given the limitations of this method, in the next section we compare it to a previously established method which uses BC as a tracer for primary particles.

### 4.3.2 BC tracer method

In this section, we used BC concentration as a tracer for estimating the number fraction of primary particles (Section 3.2). We first compare this method to the non-volatile measurements (section 4.3.1), and then to extend the analyses of primary and secondary particles to the full-year measurement period. For optimizing the BC-tracer method for Payerne, we assessed the sensitivity of choosing different percentiles (e.g. 0.1%, 0.5%, 1% and 2%) of observed N-to-BC ratios to infer the $S_1$ factor in Eq. 2. The value of $S_1$ showed low sensitivity to percentile changes, as observed in previous locations (Rodríguez and Cuevas, 2007). Since, $S_1$ is a simple factor used to estimate the number of primary particles per nanogram of BC, the low sensitivity of $S_1$ to different percentiles directly translates to high precision of the estimated primary particle fraction (Fig. S14B). This approach does not systematically misclassify the primary volatile particles as secondary particles, whereas it has its own limitations. The approach is biased towards primary particle sources with the lowest $N_{UFP}$-to-BC ratio, e.g. traffic-related particles (Rodríguez and Cuevas, 2007). A recent study estimated that the method leads to an overestimation of the primary vehicle exhaust particles concentrations by 18% and 26% in urban and suburban sites, respectively, but no impact on secondary particles (Casquero-Vera et al., 2021). Given that traffic has also a dominant contribution to BC mass at this site (Grange et al., 2020), a potential overestimation in the traffic contribution to the total primary particles is also possible in this study. However, it must be noted that the underlying assumption of a constant $N_{UFP}$-to-BC ratio will never provide accurate results in an environment where the contributions of different primary sources and the age of plumes show great diurnal and seasonal variations.

In a next step, we compared the results from both methods (Fig. 4, Fig. S15). Both methods show the best agreement when the 1st percentile of the NUFP-to-BC ratios is chosen as S1. The results shown in Fig. 4 demonstrate the general consistency between the results for the common measurement period, in winter, for all three size segregated fractions. However, despite the overall agreement, certain discrepancies remain when considering time-resolved data, e.g. in diurnal patterns and time series and are attributed to the limitations of both methods. Both approaches likely represent a low estimate of the primary particle fraction.. Our results, from both methods combined, suggest that, on average, 25% of the UFP in winter in Payerne originate from primary processes (Fig. 4), while the remaining are formed in the atmosphere following oxidation of precursor gases or chemical processing of available aerosol particles. Based on the BC tracer method, 2.15 (x $10^6$) cm$^{-3}$ primary particles are observed per 1 ng/m$^3$ of BC. Our results are in line (slightly lower given the nature of our measurement location) with previous measurements in Switzerland with 3.1 x $10^6$ and 3.6 x $10^6$ cm$^{-3}$ primary particles per 1 ng/m$^3$ of BC observed in an urban background location in Lugano and an urban location in Bern, respectively (Reche et al., 2011), although differences in vehicle fleet between locations and changes over time due to more stringent emission regulations, are expected to influence this ratio. In comparison, less primary particles per 1 ng/m$^3$ BC are found in Payerne compared to other European cities such as Milan, Barcelona and Santa Cruz de Tenerife, with values exceeding 4.5 (x $10^6$) cm$^{-3}$ primary particles per 1 ng/m$^3$ of BC (Rodríguez and Cuevas, 2007). In the next section, we extend the analyses over the full year of measurements using the BC tracer method.

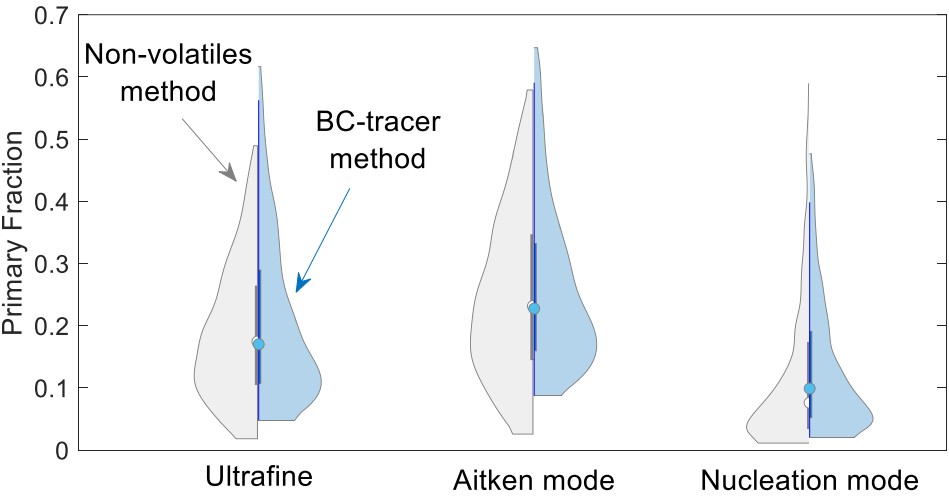

**Figure 4 Fraction of primary particles in the ultrafine, Aitken and nucleation modes using the non-volatile particle measurements in gray and using the BC tracer method in blue. Violin plots are a combination of boxplot and a kernel distribution function on each side of the boxplots. The white circles define the median of the distribution and the edges on the inner grey boxes refer to the 25th and 75th percentiles, respectively. Plots include winter hourly averages, outliers below the 5th percentile or above the 95th percentile are excluded.**

### 4.3.3 Seasonality of size-segregated primary and secondary UFPs

The longer-term results are not different from the wintertime observations: secondary particles dominate over primary ones (Fig. 5). In addition, during all seasons and for all size fractions, the higher particle concentration is

associated with a higher secondary particle concentration, demonstrating the contribution of secondary processes to increasing the number of particles in the atmosphere (Fig. S16). The primary UFPs peak in the morning and evening is concurrent with traffic rush hours, while the secondary particles show morning and evening peaks and additionally a midday peak during the spring and summer seasons (Fig. 5D). While the diurnal variability of primary UFPs is controlled by both the nucleation and Aitken mode particles, the seasonal and diurnal variability of secondary UFPs is driven by the nucleation mode particles, which show a much higher contribution and variability.

We find that the nucleation mode is dominated by secondary particles with an average fraction of 84% contribution to the nucleation mode and 47% contribution to the total UFP (Fig. 5B-C). The primary nucleation mode particles are about two times higher in concentration in spring and summer compared to the colder seasons, with two peaks related to traffic rush hours as observed for the UFPs (Rönkkö et al., 2017), consistent with increased traffic volume in summer (FEDRO, 2021). In comparison, the secondary fraction shows distinct seasonal patterns with a peak during midday only during warm seasons. Midday peaks are attributed to NPF events which are most intense during daytime – peak solar radiation (more details in section 4.4). Besides the NPF related peak during midday in warm seasons, the nucleation mode also exhibits a morning (all seasons) and evening peak (fall and winter season). The evening peak could be caused by biomass burning especially in winter (Casquero-Vera et al., 2021; Grange et al., 2020). The morning maximum of nucleation mode secondary particles occurs 1 to 2 hours after the peak of non-volatile primary traffic particles which suggests their atmospheric processing (Fig. 5F). The time lag indicates that it is not dominated by nucleation happening immediately after emission as the hot exhaust cools. Instead, it requires the accumulation of emitted vapors and/or chemical processing of these vapors.

In the Aitken mode, compared to the nucleation mode, we observe a higher contribution of primary particles, although the average fraction does not exceed 25% (Fig. 5C). In addition, while the primary Aitken mode particles display rush hour peaks, the secondary particles show a rather constant diurnal cycle (Fig. 5E). Primary Aitken mode particles originate from anthropogenic sources, i.e., traffic and residential wood burning. The secondary fraction is expected to comprise processed secondary particles emitted within a much larger spatial area (Seinfeld and Pandis, 2016).

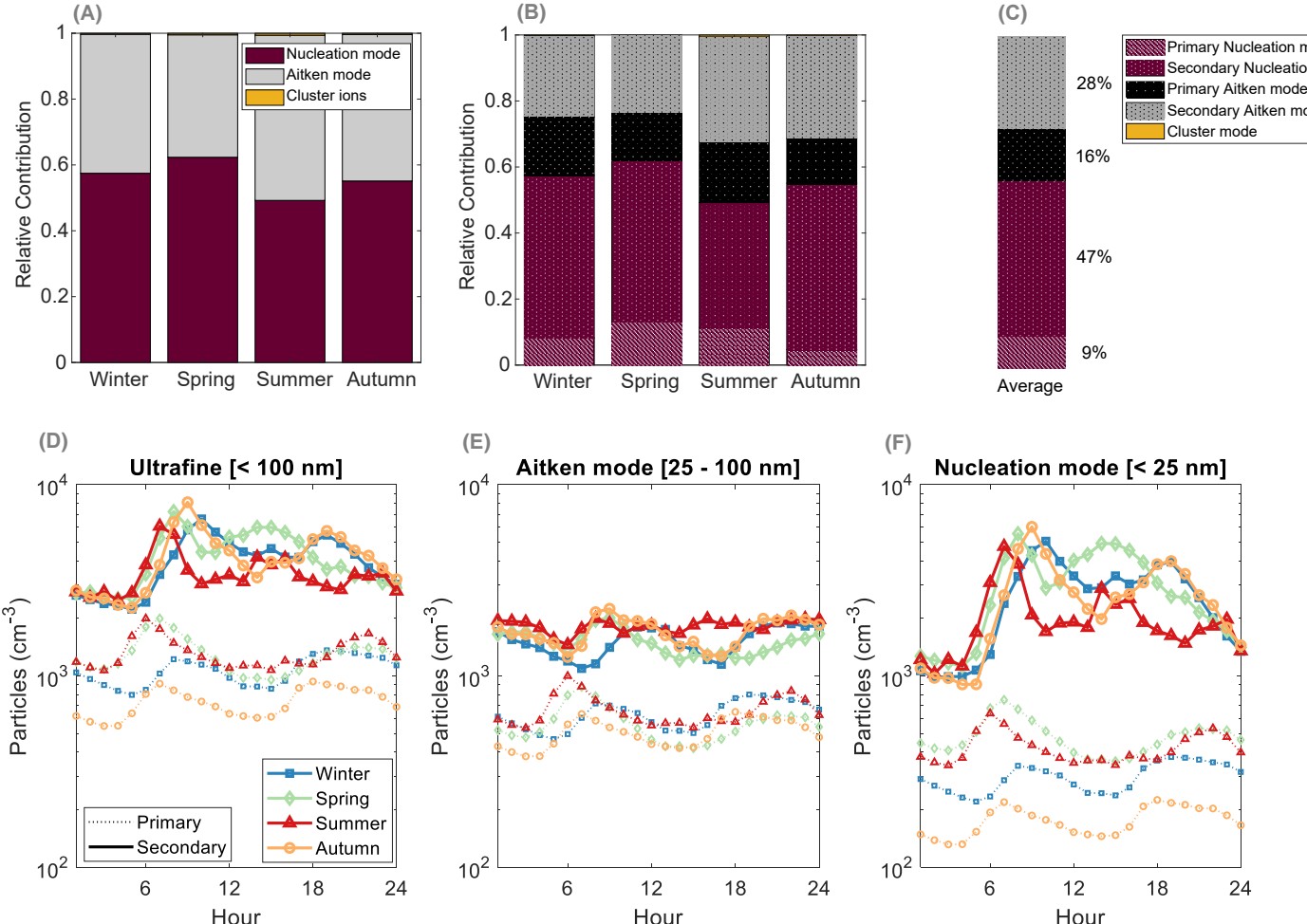

**Figure 5 (A) Relative contribution of each of the nucleation and Aitken mode particles and cluster ions to total ultrafine particle concentrations. No data from August and September are available for the nucleation mode particles or cluster ions, affecting summer and autumn means. For the Aitken mode particles, summer July includes 2020 and 2021 concentrations. Data included in nucleation mode and cluster ion mode concentrations are in 5 min time steps (NAIS time stamp), while the Aitken mode concentrations are in 3 min time steps (SMPS time stamp). (B) Similar to (A) but with the inclusion of primary and secondary fractions. (C) Similar to (B) for the yearly average. For A, B and C, the color refers to the size fraction, while the shading pattern refers to the primary and secondary fractions. (D) The particle number concentration of the primary (dashed lines) and secondary (solid lines) in the ultrafine, (E) Aitken and (F) nucleation modes diurnal averages.**

### 4.3.4 Sources of primary particles in Payerne

The BC-rich particles observed at Payerne can be attributed to wood burning and fossil fuel combustion emissions and are found in the fine mode having diameters smaller than 300 nm (Motos et al., 2020; Zotter et al., 2017; Grange et al., 2020). BC concentrations in Payerne are higher during winter compared to summer (Fig. S17 – S18), attributed to the higher contribution of wood burning for heating purposes together with a shallower boundary layer (Fig. S9). Traffic is another source of BC, which is depicted in our data as two daily peaks during rush hours: morning and evening (Fig. S17). Due to the proximity of our measurement site to the airport, BC-rich particles might be also potentially attributed to airport emissions. In Fig. S18, we assess this potential influence of airport activities on the measured BC concentrations. However, like $SO_2$, as discussed in section 4.1.2, the airport contribution to BC cannot be quantified, given the complexity of the sources around the measurement site. For instance, although the BC concentration drops by a factor of 1.3 during the daytime on weekends, this decrease could be attributed to a lower road traffic density. Additionally, BC concentrations do not decrease substantially

during airport shutdown periods (Fig. S18) and in general, wind arriving from the direction of the airport (NW: 300 – 340˚) does not show elevated BC concentrations. These observations point toward a rather low influence of the airport to our BC measurements. Our findings fall in line with Grange et al. (2020) who attributed BC concentrations observed in Payerne, over the last decade, to residential wood burning and traffic emissions.

## 4.4 Secondary UFP at Payerne: the role of new particle formation events

### 4.4.1 Frequency of NPF events

A total of 266 days were classified into the event classes (defined in Sect. 3.2) as shown in Fig. S19. Among those, 154 days were classified as nonevent constituting 57% of the total days. 54 days (20%) were classified as local NPF event days, where the particle formation is observed starting from the smallest particles (sub-3 nm cluster ions) representing local processes. On 18 days (7%), an increase in ions concentration was observed with no increase in the larger particles or obvious signs of growth, these days were referred to as ion bumps. Both local NPF and ion bump events tended to occur more frequently during the warmer months, with local NPF events even occurring on most days (~70%) in May, while nonevent days occurred more frequently during the colder months (Fig. S20). The remainder of the days which did not fit the categories of either nonevent, local NPF, or ion bump were left undefined as these were either affected by precipitation or nearby traffic – elevated $NO_2$ concentrations. We note that we did not observe any transported events at the site, without a local NPF event.

### 4.4.2 Size-segregated growth rates

The particles formed in Payerne could grow to sizes greater than a few nm in most cases (local events). The median growth rates were 1.3, 3.4 and 5.0 $nm.h^{-1}$ for the size ranges $1.5 – 3$, $3 – 7$ and $7 – 15$ nm, respectively (Fig. S19). The GRs were comparable during the seasons within the same size fraction (Fig. S21). However, although slightly higher GRs are observed in summer compared to spring, this conclusion is not certain given the low statistics (Fig. S21). Still, higher growth rates are to be expected in warmer seasons given the increased emissions of biogenic VOCs (in case the condensation sink values are similar), especially in areas surrounded by vegetation as in Payerne.

Our measured GRs in the smallest size fractions ($1.5 – 3$ and $3 – 7$ nm) are within the range of those observed in other rural environments, such as in Puy De Dome, France, (Manninen et al., 2010),Vavihill, Sweden, (Manninen et al., 2010), and the boreal forest in Hyytiälä, Finland, (Yli-Juuti et al., 2011). However, the GRs of the larger sizes exceed those observed in rural background locations and resemble more rural polluted locations such as Melpitz, Germany, and K-Puszta, Hungary, (Manninen et al., 2010). This observation might be attributed to the participation of anthropogenic sources (residential heating and highway close to Payerne) to the growth (Fig. S1) together with lower CS (median $CS_{events} = 2.6$ x $10^{-3}$ $s^{-1}$, see Fig. 6), compared to the rural polluted locations mentioned (median $CS_{events}$ (K-Puszta) $= 6.6$ x $10^{-3}$ $s^{-1}$ and $CS_{events}$ (Melpitz) $= 8.4$ x $10^{-3}$ $s^{-1}$)(Manninen et al., 2010).

### 4.4.3 Precursor vapors driving NPF at Payerne

A clear enhancement in the particle formation rate (total and ion induced) is observed on NPF event days compared to nonevent days. The median $J_{2.5}$ on event days is on average 2 times higher and during the peak up to 7 times higher than $J_{2.5}$ on nonevent days (Fig. 6A). The highest $J_{2.5}$ are observed during spring, with the highest peak occurring in March and reaching up to 3 $cm^{-3}s^{-1}$ (Fig. S22). These particle formation rates fall in the same range as other rural and rather vegetation-dominated locations such as in Hyytiälä, Finland, (Kulmala et al., 2013), Jarvseljä, Estonia, (Vana et al., 2016) and Hohenpeissenberg, Germany, (Birmili et al., 2003). The ion induced formation rates ($J_{1.5}^-$ and $J_{1.5}^+$) show a similar trend as $J_{2.5}$ peaking around noon (Fig. 6B), especially in spring (Fig. S22). Similarly, $J_{1.5}^+$ is 3 times and $J_{1.5}^-$ is 5 times on average higher on event days than on nonevent days. During peak time, $J_{1.5}^+$ is 6 times and $J_{1.5}^-$ is 12 times higher on event days than on nonevent days. The factor of 2 difference between $J_{1.5}^+$ *and* $J_{1.5}^-$ could be attributed to the composition of air (Mohnen, 1976). As for $J_{2.5}$, average $J_{1.5}^+$ and $J_{1.5}^-$ fall in the range of ion induced formation rates measured in rural locations in Europe, see Manninen et al. (2010) and references therein.

At most locations in the continental boundary layer, sulfuric acid ($H_2SO_4$) mediates NPF events, especially in the presence of fixing bases such as amines or ammonia (Yao et al., 2018; Dada et al., 2023; Almeida et al., 2013; Kürten et al., 2016; Aktypis et al., 2024). At Payerne, NPF events are observed on days with significantly enhanced of $H_2SO_4$ concentrations (calculated as a proxy from $SO_2$, see Sect. 3.4 - median events = 1.88 x $10^6$ $cm^{-3}$, median nonevents 1.07 x $10^6$ $cm^{-3}$, $p < 0.05$ using Wilcoxon's Rank-Sum and Signed-Rank tests) while $SO_2$ levels are similar on event and nonevent days (Fig. 6C, Fig. S23). Interestingly, ammonia does not show a similar behavior, but an equal median concentration and distribution regardless of whether an NPF event was observed or not (Fig. 6D). This observation allows us to conclude that while $H_2SO_4$ could be the factor controlling NPF in Payerne, ammonia is not the limiting factor. On the other hand, a clear difference between CS levels is observed on events and nonevent days, so is the case for the cloudiness (median $CS_{events}$ = 2.7 x $10^{-3}$ $s^{-1}$, $B_{events}$ = 0.72; median $CS_{nonevents}$ = 5.9 x $10^{-3}$ $s^{-1}$, $B_{nonevents}$ = 0.56) (Fig. 6E-F). The two aforementioned factors control $H_2SO_4$ formation and hence an increased CS or cloudy conditions inhibit the occurrence of NPF events. Based on these results, we can hypothesize that the limited $H_2SO_4$ formation (due to high CS or cloudiness) could be the limiting factor for NPF in Payerne.

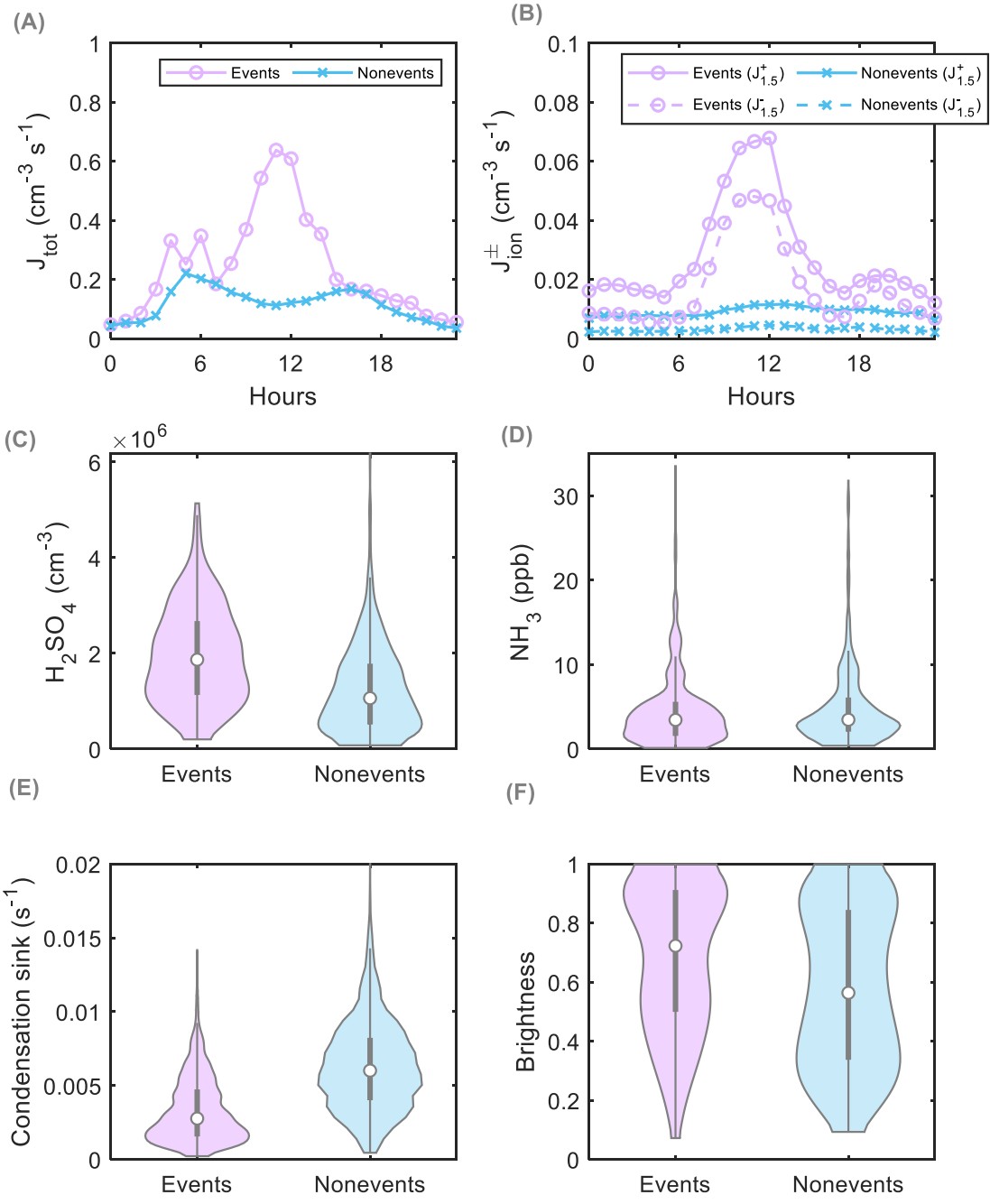

**Figure 6** (A) total ($J_{2.5}$) and (B) ion induced ($J_{1.5}$) formation rates on event (pink) and nonevent days (blue). (C) H₂SO₄ proxy concentration, (D) NH₃ concentration, (E) condensation sink and (F) brightness (clear-sky parameter) on event (pink) and nonevent days (blue) shown as violin plots. Violin plots are a combination of boxplot and a kernel distribution function on each side of the boxplots. The white circles define the median of the distribution and the edges on the inner grey boxes refer to the 25th and 75th percentiles, respectively.

To confirm our hypothesis, we plot the observed NPF probability as a function of $H_2SO_4$ and CS in Fig. 7. We find that events occurred most frequently under high $H_2SO_4$ exceeding $4 \times 10^5$ $cm^{-3}$ and CS below 0.01 $s^{-1}$ (Fig. 7). The highest probability is observed in the lower-right corner of Fig. 7, under high $H_2SO_4$ concentration and the lowest CS. As $H_2SO_4$ formation depends on the level of solar radiation, summer conditions favor $H_2SO_4$ formation, see also Fig. S24. $SO_2$ which is always available in the regional background air, has similar concentrations on event and nonevent days (Fig. S23), and higher concentrations are associated with air masses arriving from the northeast, passing over the town of Payerne (Fig. S25). During less cloudy conditions (high B), $SO_2$ oxidation is favored, producing higher concentrations of $H_2SO_4$. At the same time, cleaner conditions (low CS), favor NPF occurrence in two ways. First, the loss of $H_2SO_4$ to preexisting particles is decreased, favoring an increased lifetime of the precursor gas needed for NPF (see equation 6). Second, besides $H_2SO_4$, fewer freshly formed particles and precursor vapors needed for particle growth are lost to preexisting background particles, increasing the probability of NPF.

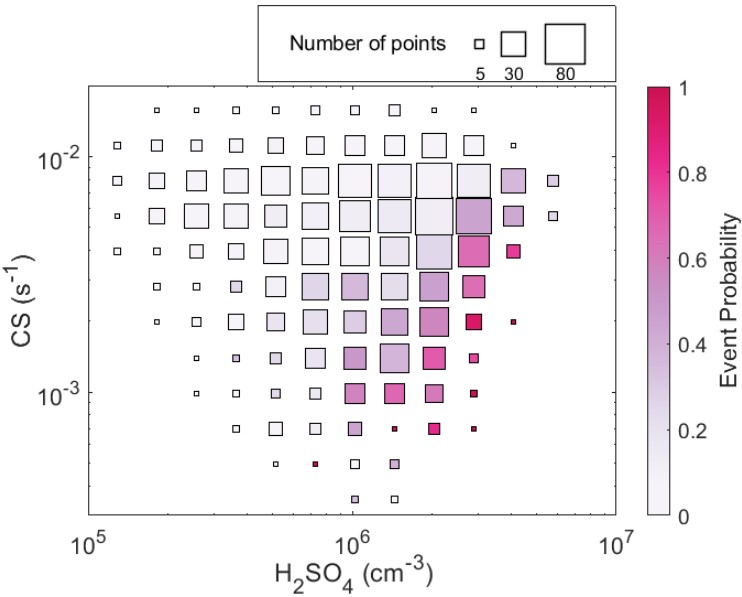

**Figure 7 Event probability distribution at Payerne based on the CS and sulfuric acid proxy hourly concentrations during daytime [8:00 – 17:00 CET]. Marker size indicates number of days included in the probability calculation within every cell.**

In Payerne, NPF events occurred most frequently when the wind arrived from the southwest areas dominated by vegetation (grasslands and croplands) and associated with a lower CS suitable for NPF events to occur (Fig. S25-S26). Besides $H_2SO_4$, stabilizing bases are needed for NPF to occur in boundary layer conditions. In Switzerland, $NH_3$ and amines originate from agricultural practices, specifically from animal housing and manure storage and spreading especially in the central and western parts (Reidy et al., 2008). Wind carried similar $NH_3$ concentrations from all directions at our measurement site, as expected given the area's proximity to agricultural areas and croplands (Fig. S25). As mentioned earlier, the $NH_3$ concentrations are also similar on NPF events and nonevent days, they are also sufficiently high to not limit NPF on days when sufficient $H_2SO_4$ is available. Although we have no amine measurements available, the wind direction carrying $NH_3$ is likely carrying amines as well. Amines

such as methylamine, dimethylamine (DMA), and trimethylamine are expected to be abundant in rural locations in the vicinity of agricultural activities (Kürten et al., 2016; Ge et al., 2011).

By comparing our observations to previous field measurements and chamber experiments, we find that the data points from Payerne fall in the middle between all other measurement sites, boreal, rural and agricultural showing the complexity of the different sources (Fig. 8). During warmer periods, our data points fall in line with the measurements from the Po Valley, during spring. The authors of the Po Valley study report $H_2SO_4$-DMA as the main mechanism of NPF in this location aided by other bases such as $NH_3$; and estimate the $NH_3$ and DMA concentrations to be 10.6 ppb and less than 10 ppt, respectively (Cai et al., 2024). Moreover, our data points fall at a slightly higher slope than those measured in an agricultural land in Finland (Qvidja – yellow circles [temperature from 5 to 25℃]) which were found to be mediated by $H_2SO_4$-$NH_3$ and extremely low volatility organic vapors of biogenic origin (Dada et al., 2023; Olin et al., 2022). Our data points fall on the same line as $H_2SO_4$-DMA line (4 ppt DMA), based on kinetic-model parametrization at 25℃ from chamber experiments and are higher than those parametrized from $H_2SO_4$-$NH_3$ experiments (Xiao et al., 2021). The scatter of the points and deviation from the exact line can be largely attributed to the variable concentration of DMA as well as the contribution of other vapors in the nucleation and growth process, e.g. biogenic and anthropogenic organics which cannot be excluded.

Under colder temperature conditions, data points in Payerne coincide with measurements from Hyytiälä, where the average $H_2SO_4$ concentrations are rather low, accompanied by a low average ammonia and DMA concentrations, of 0.066 ppb and below detection limit, respectively (Hemmilä et al., 2018). However, the CS levels are also comparably low ($CS_{events}$ = 1.4 x $10^{-3}$ s$^{-1}$), together with colder temperatures, favor NPF events at these low concentrations. Hyytiälä forest site is also surrounded by a trees belt emitting monoterpenes, further supporting particle nucleation and growth (Rinne et al., 2005). Similar to during warmer temperatures, the points from Payerne fall on the same line as $H_2SO_4$-DMA line (4 pptv DMA), based on kinetic-model parametrization at 5 ℃ from chamber experiments and at a higher slope than $H_2SO_4$-$NH_3$ chamber experiments (Xiao et al., 2021; Lehtipalo et al., 2018).

Based on these observations, we can conclude that NPF events in Payerne are mediated by $H_2SO_4$-DMA supported by ammonia, with the availability of $H_2SO_4$ (and low CS) being the decisive factor. Given the environment surrounding our measurement location, we can speculate that both biogenic and anthropogenic organic vapors contribute to the growth of the freshly formed particles. However, such conclusions can be only fully verified with additional measurements of precursor vapors.

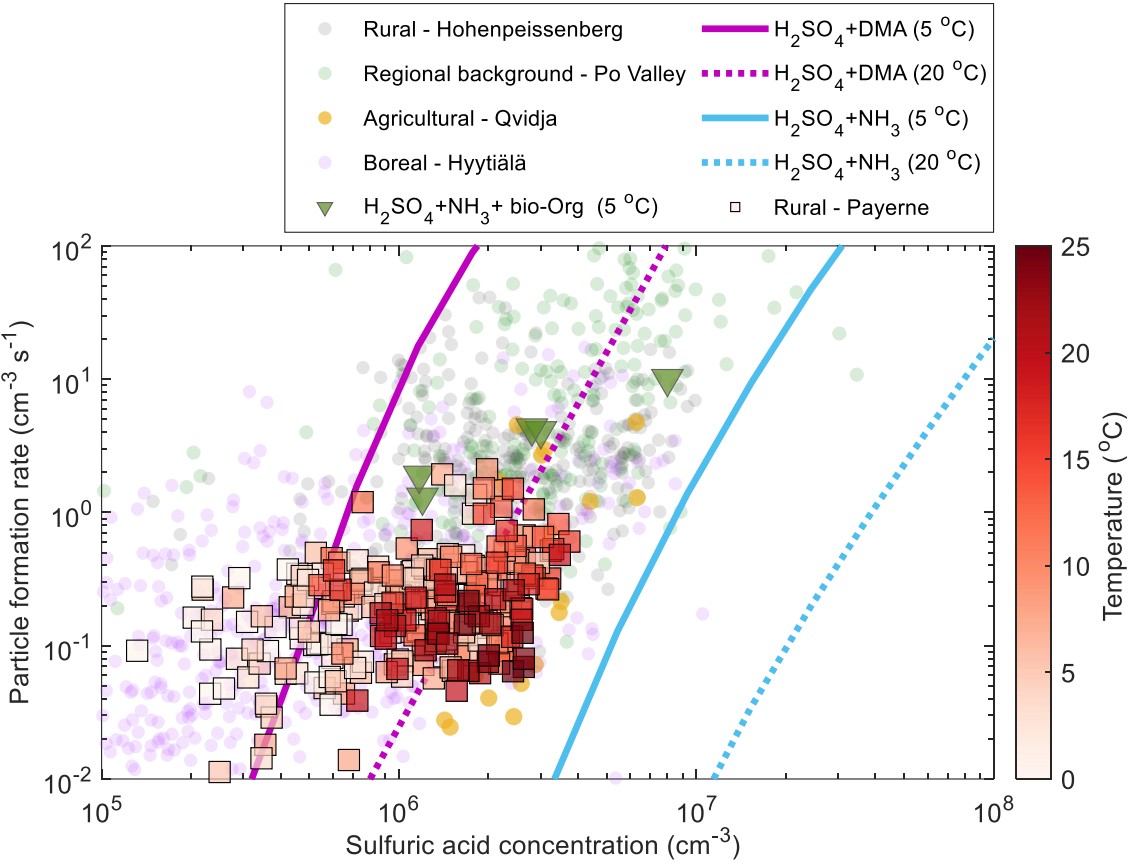

**Figure 8 Particle formation rates as a function of H₂SO₄ concentration, daily averages (8:00 – 17:00 CET) – squares, are colored with ambient temperature. Parametrizations based on chamber measurements of H₂SO₄+NH₃ (2 ppbv) ammonia at 5°C (solid cyan line) and 20°C (dotted cyan line) are shown (Xiao et al., 2021). Similarly, parametrizations based on chamber measurements of H₂SO₄ + dimethylamine (DMA – 4pptv) at 5°C (solid magenta line) and 20°C (dotted magenta line) are shown (Xiao et al., 2021). Green triangles are H₂SO₄+NH₃ (0.1 – 1 ppbv) at 5°C in the presence of constant monoterpenes (α-pinene and Δ-3-carene) and NOx from Lehtipalo et al. (2018). Filled translucent points in the background are atmospheric measurement from the boreal forest (pink), rural location in Germany (grey)(Birmili et al., 2003), a regional background in Italy (green) (Cai et al., 2024) and an agricultural land (yellow)(Dada et al., 2023). Particle formation rates from chamber measurements are at 1.7 nm while field observations are either measured at 1.5 nm or extrapolated to 1.5 nm using the Kerminen and Kulmala equation (Kerminen and Kulmala, 2002) (see section 3.7). For Payerne, H₂SO₄ concentrations are derived from proxy calculations (see section 3.8).**

### 4.4.4    Contribution of NPF particles to UFP

The analyses performed within the German Ultrafine Aerosol Network (GUAN) show that the contribution of NPF to UFPs was about 13 %, 21 %, and 7 % for the urban background, regional background, and low mountain range, respectively (Sun et al., 2024). We applied similar analyses to our data; although, our results might be subject to uncertainty due to low statistics especially during low radiation autumn or winter (Fig. S27). We found that the overall NPF contribution to the UFP was 5.07%. However, if we limit our analysis to the NPF time window (11:00 to 18:00 CEST), we find that the contribution of NPF to total UFP increased to 13.04%. Limiting our analysis further to the 3-hour peak NPF window, the contribution of NPF to total UFP increased to 20.4%. By combining the information on the secondary UFP fraction (section 4.3.3) with this analysis, we find that NPF explains an average of 6.8 % of the secondary UFP. A main reason for this low fraction is that the secondary

particles also contain a substantial portion of volatile particles peaking shortly after the traffic rush hour, which are not counted in the NPF contribution.

We also performed the same analysis on the size segregated particles and found an overall contribution of NPF to nucleation mode particles of 9.32%, up to 21.3% during the NPF time window (11:00 to 18:00 CEST) and 32.1% during the 3-hour NPF peak. These results translate to ~20% contribution of NPF to secondary nucleation mode particles. On the other hand, the overall contribution of NPF to Aitken mode particles is near 0, even during the NPF time window. Such an observation is expected, as the Aitken mode particles are rather transported particles and not formed via local processes, see also section 4.3.3.

## 5      Conclusions

Understanding the sources of UFP in rural environments is crucial for assessing air quality, public health impacts, and the influence of both natural and anthropogenic emissions on regional atmospheric chemistry. In this work, we investigate the outdoor sources of UFPs at Payerne, an ACTRIS and NABEL site in rural Switzerland. Although UFPs in rural areas have not been as extensively studied as those in urban settings, our research demonstrates that secondary processes can elevate UFP concentrations in rural environments to levels similar to those found in urban areas. First, we identify and quantify primary and secondary UFP sources using long-term particle and ion number size distribution measurements starting from ~1 nm. By combining measurements of non-volatile particles and long-term BC concentrations, used as tracers for primary particles, we find that secondary particles dominate over primary particles in all size fractions.  A higher secondary fraction is observed at higher particle concentrations demonstrating the role of atmospheric processing in increasing the number concentration of particles in the atmosphere.

Primary particles in Payerne comprise those originating from traffic and wood burning, mostly non-volatile and BC-rich. While the concentrations are relatively similar between the different seasons, the diurnal pattern shows the unique peaks concurrent with morning and evening traffic rush hours. In comparison, secondary particles, originate from NPF events and processing of primary particles. UFPs diurnal and seasonal variability is determined by the nucleation mode particles dominating in spring and summer due to local emissions and secondary aerosol formation. NPF happens in spring and summer and contributes largely to the nucleation mode (~ 20%). Based on several lines of evidence, we find that NPF events in Payerne are driven by $H_2SO_4$ and stabilizing bases likely DMA supported by ammonia, with the availability of $H_2SO_4$ in combination with sufficiently low CS being the decisive factor. Given the location and multitude of sources, it is likely that both biogenic and anthropogenic organic vapors contribute to the growth of the newly formed particles especially under favorable meteorological conditions. However, additional measurements of precursor vapors are needed to fully verify these conclusions.

## 6      Acknowledgements

We acknowledge Maxime Hervo, Alexander Häfele and Phillippe Overney from MeteoSwiss in Payerne for providing the infrastructure access and support with the measurements. We further acknowledge MeteoSwiss collaborators at Payerne for the operational SMN and EMER-Met measurements. Nicolas Bukowiecki is further acknowledged for his contribution to the fund raising of the GAW-CH science project that initiated the aerosol insitu measurements in Payerne. Günther Wehrle, Pascal Schneider and Levi Folghera are acknowledged for the technical support of the aerosol in situ observations. We also thank Kaspar R. Daellenbach for scientific discussions.

## 7      Funding

Initial pilot implementation of aerosol in situ measurements at the Payerne observatory received financial support from MeteoSwiss through a science project in the framework of the Swiss contribution to the Global Atmosphere Watch programme (GAW-CH) of the World Meteorological Organization. Further upgrades and operation of the ACTRIS observations at Payerne were funded by the Swiss State Secretariat for Education and Research and Innovation (SERI) in the framework of ACTRIS Switzerland. Further operation support was received from topical centers of the ACTRIS ERIC. L.D. acknowledges funding from the SNSF (grant number 216181). N.E. thanks cross-atmospheric research infrastructure services provided by ATMO-ACCESS (EU grant agreement No 101008004).

## 8      Author Contributions

Conceptualization: LD
Fund raising: MGB, BB
Data collection: BB, MCC, CH, MS
Data Analysis: BB, NN, LA, LD, MCC, CH, MS
Data interpretation and scientific discussions: all
Writing: LD
Review and commenting: all

## 9      Conflict of interest

The authors declare no conflict of interest.

## 10      Data availability

Global radiation, temperature, precipitation, wind speed and wind direction data can be obtained from the SwissMetNet (https://opendata.swiss/de/dataset/automatische-meteorologische-bodenmessstationen).    Trace gases data are available open access on the National Air Pollution Monitoring Network (NABEL) website

(https://www.bafu.admin.ch/bafu/en/home/topics/air/state/data/data-query-nabel.html ). All other data would be provided from the corresponding authors upon request.

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
