# Peer review of "Sources of ultrafine particles at a rural Midland site in Switzerland"

_Aerosol Research, 2024_

## Referee Comment (RC1)

**Overall quality**

I consider the overall quality of the manuscript as very good considering a suite of instruments deployed for the study and the use of appropriate data analysis tools. The authors have also carefully considered the limitations and uncertainty some specific part of the results in this work.

However, the authors need to ensure that sentences are concise and avoid unnecessary repetition and confusion in the statement formation.

**General comments**

This research investigates the sources of ultrafine particles (UFPs) in rural Payerne, Switzerland. It finds that secondary processes significantly elevate UFP concentrations, similar to urban levels. Primary particles mainly come from traffic and wood burning, while secondary particles arise from new particle formation (NPF) events, driven by sulfuric acid and stabilizing bases like DMA and ammonia. The study highlights the need for further measurements of precursor vapours to fully understand these processes. Considering the dearth of studies of UFP in rural locations, this study adds to the research enhancement in the field of particle formation.

I recommend the publishing of this work after the author addresses the specific comments and also clarifies the readability of the text. This MS needs English language revision, many sentences are poorly formed and fail to convey the correct meaning.

**Specific comments**

Line 20: non-volatile particles fraction should be written as non-volatile fraction of particles, to be corrected else where in the MS.

Line 23: Expand the acronym NPF. the acronyms need to be defined at their first use.

Line 24: What do you mean by cluster ions and nucleation mode. The size range needs to be defined in the abstract itself.

Line 30 " transport related" should be replaced by "traffic related". Transport is a verb – a process of moving goods/people from one place to another.

Line 49-51: abundant references are available. The authors need to cite some of those.

Line 58: The authors talk about that different sources can contribute to different size classes of UFPs. Have the authors defined what are the important size classes of UFPs relevant here?

Line 89 Figure S1: Not clear at all. the military airport is not clearly visible, Image resolution needs to be improved. Also I think the authors meant fig S1(A)?

Line 113-114 : Can the authors give a range of overestimation in MLH expected. Using the word "slight overestimation" does not sound

Line 129: The word "particle" can be omitted from number size distribution, as here the authors are talking about the ions.

Line 131. What do the authors mean by "multiply charged particles"? is it a typo error?

Line 149: SMPS 3938 size range in the table S1 is mentioned as 6-110nm, here it is 3-110 nm.

Line 187 -188: regional events are also well defined in Dal Maso et al. The authors mentioned they followed Dal Maso classification and later modified to include regional &t transported events as per Dada et al.. what was the modification used in the classification of regional events here and why?

The later lines in the same paragraph explain this better. However, that means Dal Maso et al classification was not modified as mentioned in the Lines 187-188 but Dada et al classification was adapted to fit the classification at the study site (as per the figure S3). The authors are suggested to modify the text to bring more clarity in the classification explanation as this an important section in the MS.

Section 3.4 : Coagulation sink was calculated using the combined size distribution method. The authors should include the main equation under this section or atleast in the supplementary information.

Line 200: Why was 50% appearance time method used "positive ions" only?

Line 281: "Traffic emissions are a major source of NO2 indicated by the sharp increase of NO2 during the morning and evening rush hour (Fig. 1, Fig. S7), although other sources such as residential wood burning and use of fertilizers in agriculture could affect the concentration". The authors attribute the morning and evening peaks mainly to the traffic rush hours. First for a rural place would the evening rush hours be 18-20 hrs? do we have data for traffic rush hours? Have the authors checked the BC concentration during these hours? do they match with traffic hours?  Could there be other nighttime chemistry playing a role for NO2 here? Significant amount of $NO_2$ can reacts to form $NO_3$ and $N_2O_5$ during the course of a night, but their fate is an important determining factor to the overall fate of $NO_x$ (=NO and $NO_2$)

Line 284 : Figure S8 B just shows the NO2 concentration on weekdays between 8-17 hrs, which does not justify the high concentration during 18-20 hrs.

Line 285-286: Mixing layer height does not clearly explain the seasonal differences in NO2 concentrations. The MLH is highest in spring, yet NO 2 concentration is higher as compared to summer season, so the authors need to explain other processes associated with seasonal variability of NO2.

Line 308 "regional nature of SO2 emissions": Could the authors be more explicit in explaining this ? Since the airport emissions are not playing a significant role, then which sources are impacting the SO2 emissions is missing from the discussion.

Line 363: Does the catalytic stripper separates the semi volatiles or non-volatiles? How efficient is this separation? is there any error estimates using this technique, since the results of secondary particles may itself be at high uncertainty as the authors have already mentioned. therefore, the result uncertainty needs to be calculated atleast based on the instrument/technique used.

Line 442: it points towards the higher significance of secondary particle contribution towards total number of particles. Therefore the fig S15, y axis should have secondary fraction. I suggest to show secondary fraction on the y-axis.

Line 459 "Volatility and size suggest a secondary processing"- A vague sentence. The authors can make it more clear and concise.

Line 518-519: In the previous lines the authors stated that airport is not making a big difference in the anthropogenic contribution at the study site. Now in these lines, the authors mention airport as one of the anthropogenic sources for higher GR in the larger sizes. it is a contradicting statement! authors need to clarify.

Line 490: when the wind is coming from the airport direction then also the BC concentration is not elevated! What could be the reason for this? BC concentration should normally go high when the wind is from a source of BC. Mixing of airmass before arrival at the site? This needs to be explained here.

Line 541 -542: An interesting observation! Can the authors comment what could be the reasons for a similar median ammonia concentration during NPF and non NPF days? why ammonia is not playing a limiting role despite the importance of a base, which could play an important role in NPF (stabilizing SA clusters) as per many previous NPF studies! Is it because of low ammonia threshold?

Line 557: "causal relationship"? please refine the sentence for better clarity.

Line 563-564: If the precursor vapours required for growth are lost (for e.g organics) how does it favour NPF? The manuscript does treat NPF as clustering + growth.

Line 604: Are the authors sure that the NPF events are supported by ammonia based on the fig 6D where ammonia conc. is same during the event and non-event day.

Line 647-648: "secondary particles dominate the overall particle size distribution, with a larger fraction at higher particle concentrations" could be streamlined for clarity.

Line 666: The conclusion section of the MS should also mention about the role of meteorology (wind direction) in driving NPF especially considering the location of the site surrounded by grasslands and agriculture lands.

**Technical comments**

Table S1, two times "nm" is mentioned in the SMPS row, delete one.

Line 86-88: sometimes "Km" is used for distance, sometimes its expanded form "kilometers", same for meters. Please follow consistency as per the journal guidelines.

---

## Author Comment (AC1)

**Responses to reviewer comments for manuscript ar-2024-35 entitled: "Sources of ultrafine particles at a rural Midland site in Switzerland"**

We thank all three reviewers for their thorough review and useful comments. We address their comments and suggestions point-by-point in the document below. Text by the reviewers is shown in black, responses in blue and additions to the manuscript are in orange.

**Reviewer # 1**

**Overall quality**

I consider the overall quality of the manuscript as very good considering a suite of instruments deployed for the study and the use of appropriate data analysis tools. The authors have also carefully considered the limitations and uncertainty some specific part of the results in this work.

However, the authors need to ensure that sentences are concise and avoid unnecessary repetition and confusion in the statement formation.

**General comments**

This research investigates the sources of ultrafine particles (UFPs) in rural Payerne, Switzerland. It finds that secondary processes significantly elevate UFP concentrations, similar to urban levels. Primary particles mainly come from traffic and wood burning, while secondary particles arise from new particle formation (NPF) events, driven by sulfuric acid and stabilizing bases like DMA and ammonia. The study highlights the need for further measurements of precursor vapours to fully understand these processes. Considering the dearth of studies of UFP in rural locations, this study adds to the research enhancement in the field of particle formation.

I recommend the publishing of this work after the author addresses the specific comments and also clarifies the readability of the text. This MS needs English language revision, many sentences are poorly formed and fail to convey the correct meaning.

We thank the reviewer for these comments and suggestions. We address them point-by-point in the document below and review the overall manuscript for repetitions as recommended.

**Specific comments**

1. Line 20: non-volatile particles fraction should be written as non-volatile fraction of particles, to be corrected else where in the MS.

Thank you. Modified.

2. Line 23: Expand the acronym NPF. the acronyms need to be defined at their first use.

Thank you. Added.

3. Line 24: What do you mean by cluster ions and nucleation mode. The size range needs to be defined in the abstract itself.

Added.

4. Line 30 " transport related" should be replaced by "traffic related". Transport is a verb – a process of moving goods/people from one place to another.

Modified.

5. Line 49-51: abundant references are available. The authors need to cite some of those.

Relevant references are added.

6. Line 58: The authors talk about that different sources can contribute to different size classes of UFPs. Have the authors defined what are the important size classes of UFPs relevant here?

Added.

7. Line 89 Figure S1: Not clear at all. the military airport is not clearly visible, Image resolution needs to be improved. Also I think the authors meant fig S1(A)?

We thank the reviewer for this comment. We modified Fig. S1 to increase its resolution to the maximum possible, and included a link address which allows for a user-defined magnification and navigation at the site. We also marked the location of the airport on the map. The text is modified as Fig. S1A as well as the figure caption as follows:

*Figure S 1 (A) Location and (B) land use information of Payerne measurement station. Measurement station and near-by military airport are marked with a cyan and a black pin, respectively. Maps are extracted from ESA WorldCover © (https://bit.ly/3Q6v40v, last access 06.02.2025).*

8. Line 113-114 : Can the authors give a range of overestimation in MLH expected. Using the word "slight overestimation" does not sound

A comparison between the MLH during night estimated by the bulk Richardson method and that estimated by the T inversion or by the parcel method leads to a maximum of 200 m overestimation by the bulk Richardson method.

We added this information to the text:

*During nighttime, when the MLH is lower than the first level of measurement of the MWR, the data are unavailable resulting in a slight overestimation (a maximum of 200 m) of the average MLH.*

9. Line 129: The word "particle" can be omitted from number size distribution, as here the authors are talking about the ions.

Thank you. Modified.

10. Line 131. What do the authors mean by "multiply charged particles"? is it a typo error?

The SMPS data inversion software accounted for multiple charged particles i.e., doubly, triply, and quadruply charged particles, we clarified this in the text as follows:

*The SMPS data inversion software accounted for multiple charged particles. Furthermore, a size dependent correction factor for the CPC counting efficiency and particle losses within the*

*instrument and inlet lines was applied to the data (Liscinsky and Hollick, 2010; Yook and Pui, 2005).*

11. Line 149: SMPS 3938 size range in the table S1 is mentioned as 6-110nm, here it is 3-110 nm.

We apologize for the typo; we modified the text accordingly.

12. Line 187 -188: regional events are also well defined in Dal Maso et al. The authors mentioned they followed Dal Maso classification and later modified to include regional &t transported events as per Dada et al.. what was the modification used in the classification of regional events here and why? The later lines in the same paragraph explain this better. However, that means Dal Maso et al classification was not modified as mentioned in the Lines 187-188 but Dada et al classification was adapted to fit the classification at the study site (as per the figure S3). The authors are suggested to modify the text to bring more clarity in the classification explanation as this an important section in the MS.

Dal Maso et al.'s method was modified by Dada et al. (2018) to include sub-3 nm size distribution allowing for the distinction between local and regional events. We modified the text as follows to ensure more clarity:

*Days in Payerne were classified into NPF event or nonevent days depending on the evolution of their particle number size distributions. Here, we followed the traditional method introduced by Dal Maso et al. (2005), and modified by Dada et al. (2018) to distinguish between local and transported events based on the size distributions below 3 nm. Here, we combined both aforementioned methods yet tailored them to fit our measurement location better, which is subject to traffic emissions, as the previous two methods were developed for the boreal forest environment.*

13. Section 3.4: Coagulation sink was calculated using the combined size distribution method. The authors should include the main equation under this section or at least in the supplementary information.

The equations for calculating the condensation and coagulation sinks are now included in the supplementary information.

14. Line 200: Why was 50% appearance time method used "positive ions" only?

We choose to calculate the apparent growth rate from ions which allow access to the size distribution below 3 nm. In a location such as Payerne, where the dominant nucleation mechanism is neutral $H_2SO_4$-amine clustering, the transition from charged to neutral is very short resulting in little to no difference between the growth rates retrieved from charged and those retrieved from total (charged + neutral) particles (Huang et al., 2022; Gonzalez Carracedo et al., 2022). The choice of positive over negative ions was done based on the higher observed concentration of positive ions (Fig. R1). While the difference between both polarities is attributed to the composition of air (Mohnen, 1976), previous research suggests that the ion induced NPF processes are more prominent in the positive channel (Baalbaki et al., 2021; Bianchi et al., 2016).

The text now reads:

*Positive ions were chosen for the GR calculation as those have been found more important than the negative ion when it comes to ion induced nucleation from biogenic precursors (Baalbaki et al., 2021; Bianchi et al., 2021). In addition, in a location such as Payerne, where the dominant nucleation mechanism is neutral $H_2SO_4$-amine clustering (shown later in section 4.4.3), the transition from charged to neutral clusters is very short resulting in little to no difference between the growth rates retrieved from charged and those retrieved from total (charged + neutral) particles (Huang et al., 2022; Gonzalez Carracedo et al., 2022).*

[Figure]

*Figure R 1 Comparison between positive and negative ion concentrations shown as (A) hourly and (B) daily averages.*

15. Line 281: "Traffic emissions are a major source of NO2 indicated by the sharp increase of NO2 during the morning and evening rush hour (Fig. 1, Fig. S7), although other sources such as residential wood burning and use of fertilizers in agriculture could affect the concentration". The authors attribute the morning and evening peaks mainly to the traffic rush hours. First for a rural place would the evening rush hours be 18-20 hrs? do we have data for traffic rush hours? Have the authors checked the BC concentration during these hours? do they match with traffic hours? Could there be other nighttime chemistry playing a role for NO2 here? Significant amount of $NO_2$ can reacts to form $NO_3$ and $N_2O_5$ during the course of a night, but their fate is an important determining factor to the overall fate of $NO_x$ (=NO and $NO_2$)

Line 284 : Figure S8 B just shows the NO2 concentration on weekdays between 8-17 hrs, which does not justify the high concentration during 18-20 hrs.

Line 285-286: Mixing layer height does not clearly explain the seasonal differences in NO2 concentrations. The MLH is highest in spring, yet NO 2 concentration is higher as compared to summer season, so the authors need to explain other processes associated with seasonal variability of NO2.

The reviewer is right, the $NO_2$ diurnal and seasonal variabilities are likely affected by several factors, including rush hours, multiple emission sources and nighttime chemistry. First, regarding rush hours, we compare to the seasonal BC concentrations (please refer to Fig. S16) as well as BC from traffic derived by Grange et al. (2020). Peak BC, specifically traffic-related BC, appears between 18:00 and 20:00 local time (Fig. R2). The afternoon rush hour peak for $NO_2$ is basically absent in spring and summer. This can be explained by the later sunset during the warm seasons and the correspondingly delayed onset of the development of the nocturnal boundary layer after the end of the rush hour traffic peak.

Regarding the seasonality, the BC seasonality is different than that of $NO_2$, where BC shows similar diurnal behavior in summer and spring while $NO_2$ shows higher concentrations in spring regardless of the growing boundary layer height after winter. Previous measurements in Payerne and Taenikon, another rural location in Switzerland, show lowest $NO_2$ concentrations in summer and attributed this observation to higher temperatures, increased photochemistry, lower wind speed and reduced emission sources in summer compared to the rest of the seasons (Steinbacher et al., 2007). Overall, the observed $NO_2$ diurnal cycles (and their seasonality) are a result of a complex interplay of emissions, meteorology, like dilution through advection and convection, (photo-)chemical processes during day and night, as well as removal processes such as the wet and dry deposition of $HNO_3$. An extended discussion of all these processes goes the beyond the scope of this manuscript and would also be highly speculative due to the sparse observational foundation apart from the continuous $NO_2$ measurements.

We modified the text related to the $NO_2$ concentrations – as suggested by reviewers 1 and 3 - as follows:

*Traffic emissions are a major source of $NO_2$ indicated by the sharp increase of $NO_2$ during the morning and evening rush hour (Fig. 1, Fig. S7), although other sources such as residential wood burning and use of fertilizers in agriculture could affect the concentration (Jion et al., 2023). $NO_2$ is observed in higher concentrations on workdays compared to weekends in line with the higher traffic volumes on workdays (Fig. S8B), further confirming the major contribution of traffic emissions to $NO_2$ concentration in Payerne. The variation between the different seasons is attributed to a shallower boundary layer in winter leading to accumulation of $NO_2$, compared to better dispersion and vertical mixing during summer (Fig. S9). The afternoon rush hour peak for $NO_2$ is basically absent in spring and summer. This can be explained by the later sunset during the warm seasons and the correspondingly delayed onset of the development of the nocturnal boundary layer after the end of the rush hour traffic peak.*

[Figure]

*Figure R 2 From Grange et al. (2020). Mean hourly equivalent black carbon (EBC) components for the six monitoring sites. Note the different scales on the y axes. CI: confidence interval, TR: traffic, WB: wood burning.*

16. Line 308 "regional nature of SO2 emissions": Could the authors be more explicit in explaining this ? Since the airport emissions are not playing a significant role, then which sources are impacting the SO2 emissions is missing from the discussion.

Here, we refer to $SO_2$ emissions present in the atmosphere arriving from nearby locations in Switzerland but also via long range transport from Europe in general. $SO_2$ is mostly the result of energy use and supply, through combustion of fuels containing sulfur (89%) (EEA, 2023). For clarity, we modified the text as follows:

*Overall, these observations point towards other sources (not airport related) of $SO_2$ emissions, mostly related to energy use and supply (EEA, 2023).*

17. Line 363: Does the catalytic stripper separates the semi volatiles or non-volatiles? How efficient is this separation? is there any error estimates using this technique, since the results of secondary particles may itself be at high uncertainty as the authors have already mentioned. therefore, the result uncertainty needs to be calculated at least based on the instrument/technique used.

Catalytic strippers remove the volatile fraction of an aerosol, leaving behind what is typically called the "non-volatile" or "solid" particles. The design of the instrument used (Catalytic Instrument CS015) is based on a design widely used in emissions measurements in the automotive industry, where strict requirements exist for volatile removal to report solid particle

numbers. This instrument can remove more than 99.9% of 30 nm tetracontane ($CH_3(CH_2)_{38}CH_3$) particles at inlet mass concentrations up to 1 mg/m³ (Andersson et al., 2007). While such extreme exhaust conditions - characterized by very low volatile material and high particle concentrations - are not expected in Payerne, the volatile removal efficiency is likely even higher for the data we report.

However, under specific conditions (such as high sulfuric acid concentrations combined with a high organic load), re-nucleation of particles could occur downstream of the instrument. While this is a possibility, it is unlikely in our case, and the non-volatile size distributions do not show any evidence of re-nucleation.

Although the catalytic stripper technique does have some systematic uncertainties, particularly associated with high particle losses at small sizes, the primary concern we see here is the challenge of separating primary particles from secondary particles in the volatile fraction.

We now include more details about the instrument in the methods section:

*An additional SMPS (TSI3938) was placed behind a catalytic stripper to measure the non-volatile particles in the size range 6 – 110 nm, in 1 minute time resolution. The catalytic stripper (CS015, Catalytic Instruments GmbH) has an operating gas temperature of 350°C which evaporates the volatile particles allowing the fraction of non-volatile particles to be measured by the SMPS. The design of the instrument used (Catalytic Instrument CS015) is based on a design widely used in emissions measurements in the automotive industry, where strict requirements exist for volatile removal to report solid particle numbers. This instrument can remove more than 99.9% of 30 nm tetracontane ($CH_3(CH_2)_{38}CH_3$) particles at inlet mass concentrations up to 1 mg/m³ (Andersson et al., 2007). While such extreme exhaust conditions - characterized by very low volatile material and high particle concentrations - are not observed in Payerne, the volatile removal efficiency is expected to be even higher for the data we report.*

18. Line 442: it points towards the higher significance of secondary particle contribution towards total number of particles. Therefore the fig S15, y axis should have secondary fraction. I suggest to show secondary fraction on the y-axis.

If it is acceptable by the reviewer, we would rather keep the figure as is since using both methods, we measure the primary fraction and infer the secondary one by subtraction, and to be consistent with all other figures (Fig. 4, and S13) as well as with previous literature who report similar observations. The secondary particle concentration is shown as seasonal diurnal cycle in Fig. 5.

19. Line 459 "Volatility and size suggest a secondary processing"- A vague sentence. The authors can make it more clear and concise.

Modified. The text now reads:

*The morning maximum of nucleation mode secondary particles occurs 1 to 2 hours after the peak of non-volatile primary traffic particles which suggests their atmospheric processing (Fig. 5F). The time lag indicates that it is not dominated by nucleation happening immediately after emission as the hot exhaust cools. Instead, it requires the accumulation of emitted vapors and/or chemical processing of these vapors.*

20. Line 518-519: In the previous lines the authors stated that airport is not making a big difference in the anthropogenic contribution at the study site. Now in these lines, the authors mention airport as one of the anthropogenic sources for higher GR in the larger sizes. it is a contradicting statement! authors need to clarify.

We thank the reviewer for pointing out this contradictory statement. The airport was left in the text by accident from a previous draft version of the manuscript (before we fully analyzed potential sources at the site). We therefore deleted the word "airport", and the sentence now reads:

*This observation might be attributed to the participation of anthropogenic sources (residential heating and highway close to Payerne) to the growth (Fig. S1) …*

21. Line 490: when the wind is coming from the airport direction then also the BC concentration is not elevated! What could be the reason for this? BC concentration should normally go high when the wind is from a source of BC. Mixing of airmass before arrival at the site? This needs to be explained here.

Not necessarily: while aviation gas turbine engines are a massive source of non-volatile soot number concentrations, their BC mass emissions are typically not very significant due to the small particle sizes emitted (Stacey, 2019). However, we do not see evidence of this in our data in both BC mass and non-volatile particle number concentrations when we should have clear influence based on the wind direction. We explain in the text that likely the closer and stronger sources near the sampling site obscure a clear signal and that this finding is in line with previous work from Grange et al. (2020). To further clarify this, we improved the wording of the paragraph.

22. Line 541 -542: An interesting observation! Can the authors comment what could be the reasons for a similar median ammonia concentration during NPF and non NPF days? why ammonia is not playing a limiting role despite the importance of a base, which could play an important role in NPF (stabilizing SA clusters) as per many previous NPF studies! Is it because of low ammonia threshold?

Given the temperature range in Payerne and the available $H_2SO_4$ concentrations, the observed formation rates cannot be explained by the availability of $NH_3$ alone. In Fig. 8, we show that amines, likely dimethyl amine, participate in the early stages of NPF in Payerne. Unfortunately, we do not have amines measurements, and thus cannot infer their species or exact concentrations.

23. Line 557: "causal relationship"? please refine the sentence for better clarity.

Sentence is now modified to:

*As $H_2SO_4$ formation depends on the level of solar radiation, summer conditions favor $H_2SO_4$ formation, see also Fig. S23.*

24. Line 563-564: If the precursor vapours required for growth are lost (for e.g organics) how does it favour NPF? The manuscript does treat NPF as clustering + growth.

Indeed, a lower condensation sink results in a higher availability of precursor vapors (both organics and sulfuric acid) to form and grow the particles.

The sentence now reads as:

*Second, besides $H_2SO_4$, fewer freshly formed particles and precursor vapors needed for particle growth are lost to preexisting background particles, increasing the probability of NPF.*

25. Line 604: Are the authors sure that the NPF events are supported by ammonia based on the fig 6D where ammonia conc. is same during the event and non-event day.

Given the temperature range in Payerne and the available $H_2SO_4$ concentrations, the observed formation rates cannot be explained by the availability of $NH_3$ alone. In Fig. 8, we show that amines, likely dimethyl amine, participate in the early stages of NPF in Payerne. Unfortunately, we do not have amines measurements, and thus cannot infer their species or exact concentrations.

26. Line 647-648: "secondary particles dominate the overall particle size distribution, with a larger fraction at higher particle concentrations" could be streamlined for clarity.

Sentence modified:

*By combining measurements of non-volatile particles and long-term BC concentrations, used as tracers for primary particles, we find that secondary particles dominate over primary particles in all size fractions. A higher secondary fraction is observed at higher particle concentrations demonstrating the role of atmospheric processing in increasing the number concentration of particles in the atmosphere.*

27. Line 666: The conclusion section of the MS should also mention about the role of meteorology (wind direction) in driving NPF especially considering the location of the site surrounded by grasslands and agriculture lands.

Added.

**Technical comments**

28. Table S1, two times "nm" is mentioned in the SMPS row, delete one.

Removed.

29. Line 86-88: sometimes "Km" is used for distance, sometimes its expanded form "kilometers", same for meters. Please follow consistency as per the journal guidelines.

Thank you for pointing this out. The notion in the text is now unified, for kilometers, meters and nanometers.

**Reviewer # 2**

The manuscript by L. Dada and Coauthors provides a comprehensive analysis of the phenomenology and sources of ultrafine particles at a rural site in Switzerland. The study is supported by an in-depth analysis of the seasonality and diurnal cycles of meteo parameters, major trace gases, black carbon concentrations, as well as about new-particle formation events frequency and classification. All is nicely described in the main text and in the supplementary material (containing 26 additional plots).

The main science objectives is the assessment of the primary and secondary source fractions of total ultrafine (UFP), Aitken and nucleation particles. To this aim, the traditional BC method is employed, and compared to an innovative methodology based on thermo-denuded SMPS measurements, although the latter was deployed only for a short period of time. The Authors claim that consistent results were obtained by the two methodologies. They also acknowledge that both tend to produce low estimates for the primary fraction of the aerosol (lines 419 – 420). This is a little bit in contradiction with the Authors' statement at lines 414-415: "we do not expect large uncertainties in our estimation of the primary particle contribution". In this reviewer's opinion, several low biases can arise from the BC method. The assumption about a constant N-to-BC ratio cannot hold in an environment where the contributions of different primary sources (biomass burning, traffic) vary during the day, as well as the contributions from transport vs fresh emissions. While BC mass is conserved during transport, N is not, as a consequence of coagulation processes. The choice of selecting a small N-to-BC ratio (Fig. S14A) as characteristic for primary particles guarantees non-negative fractions for the derived secondary particle concentrations, however it can lead to greatly underestimated concentrations of primary particles in conditions when N is high per unit of BC mass emitted. Fig. 2 shows that nucleation mode particle concentrations increase from 1000 cm-3 at nighttime to 5000-6000 cm-3 during the morning rush hour. According to the BC-method analysis, ca. 80 - 90% of such growth is accounted for by secondary particles (Fig. 5F), which is somewhat counterintuitive. It is true, as noticed by the Authors, that the peak in the secondary particles is delayed with respect to that in the primary fraction witnessing the occurrence of traffic-related secondary aerosols. Nevertheless, the very dominant contribution of secondary particles with respect to the primary one during the full evolution of the rush hour peak (Fig. 5F) is unexpected: not necessarily wrong, certainly quite noticeable. The BC method seems to do a better job in attributing the rush hour peak concentrations to primary particles in the Aitken mode fraction, leaving a flat diurnal profile for the secondary particle concentrations. However, the similar apportionment into primary and secondary fractions in Aitken mode aerosols between the cold and the summer season is also unexpected, because NPF is certainly much more prominent in spring-summer and should have some effect also in the large background particles range, and especially because biomass burning is largely reduced in the summer. Again, also this result is unexpected but necessarily wrong. In general, I think that Abstract and Conclusions do a bad job in highlighting the most controversial and innovative findings of this study and should be improved.

I suggest including a figure and a brief discussion about the diurnal trends of primary vs secondary UFP fractions during an average winter day, based on the results of the non-volatile particle method.

We thank the reviewer for their comments and suggestions, we included the suggested figure and discussions, comparing both methods directly and reviewed our abstract and conclusion accordingly.

First, we improved the readability of the sentences in lines 414 – 420 mentioned by the reviewer. In the first part, we meant to refer to the contribution of traffic particles to total primary particles. In the case of Payerne, which is traffic dominated compared to other primary sources such as wood burning (Fig. R2), such overestimation is not expected. Yet, an overall underestimation of primary particles is expected by both methods.

*Given that traffic has also a dominant contribution to BC mass at this site (Grange et al., 2020), a potential overestimation in the traffic contribution to the total primary particles is also possible in this study. However, it must be noted that the underlying assumption of a constant $N_{UFP}$-to-BC ratio will never provide exact results in an environment where the contributions of different primary sources and the age of plumes show great diurnal and seasonal variations.*

In a next step, we clarify the definition of primary and secondary particles in section 4.3.1 and clearly state that in our case primary particles are those with a primary, non-volatile 'core', e.g. BC while the secondary particles are all other particles, i.e. all nucleated particles are defined as secondary, no matter whether chemistry was involved prior to nucleation. While the non-volatile method is fully consistent with this definition, it does not fully hold for the BC tracer method.

As suggested by the reviewer, we also compare the primary and secondary ultrafine particle concentrations on a typical winter day for the two methods (Fig. R4). The comparison of the primary number concentration of UFP for the common period in winter 19.12.2020 to 02.03.2021, shows high a correlation with a ratio of 0.85 ± 1.05 (Fig. R5). As mentioned by the reviewer, both methods here are not touching on the semi-volatile particles or primary organic aerosols but focus on the non-volatile core of the particles. Certain discrepancy as e.g. the spikes in the non-volatiles' method in Fig. R5 can also be attributed to the incapability of the BC method to capture the short bursts in primary particles when a source (e.g. tractor) is in very close proximity to the station. The text in section 4.3.2 is modified for clarity.

*In a next step, we compared the results from both methods (Fig. 4, Fig. S15). Both methods show the best agreement when the 1ˢᵗ percentile of the $N_{UFP}$-to-BC ratios is chosen as $S_1$. The results shown in Fig. 4 demonstrate the general consistency between the results for the common measurement period, in winter, for all three size segregated fractions. However, despite the overall agreement, certain discrepancies remain when considering time-resolved data, e.g. in diurnal patterns and time series and are attributed to the limitations of both methods. Both approaches likely represent a low estimate of the primary particle fraction.*

[Figure]

*Figure R 3 Concentrations in the nucleation mode used for deriving particle concentrations from both methods. The number concentrations are underestimated by 0.72 +/- 0.21 due to difference in the cutoff diameter of the SMPS used for each of the methods, here 6 – 100 and 2.5 - 25 nm for the non-volatiles method and BC-tracer method, respectively.*

[Figure]

*Figure R 4 (A) Primary particle number concentration, for the time period 19.12.2020 to 02.03.2021 using both methods. (B) diurnal averages particle number concentration of the non-volatile/secondary (dashed lines) and volatile/primary (solid lines) in the ultrafine modes derived from the BC-tracer method (blue) and non-volatiles method (orange).*

**Minor comments:**

The x axis is missing in Fig S13.

The title of the x axis is missing in Fig S14(A).

Fig. S18 (A) lack of both x and y axes.

We apologize for this. There apparently was a problem with the files upload to the submission system. The original figures are as shown below. We will ensure a better quality of the supplementary material.

[Figure]

*Figure S 2 Non-volatiles method: Average non-volatile (primary) and secondary ultrafine particles (<100 nm) shown as bar plot. The right axis shows the fraction of primary fraction as diamonds with hollow markers indicating NPF event days.*

[Figure]

*Figure S 3 BC tracer method (A) Ultrafine particle number concentration as a function of BC. The solid lines represent specific percentiles of the UFP number to BC mass ratio distribution, i.e. the specified fraction of data points falls below these lines. For example, the first percentile corresponds to 2.15 ($x10^6$) $cm^{-3}$ primary particles per 1 $ng/m^3$ of BC are observed. (B) Relative contribution of primary and secondary ultrafine particles at different concentrations obtained from binned constrained fits from the different percentiles of N to BC ratio in Fig. S14 A. The uncertainty range indicates sensitivity of the BC tracer method to choosing the percentile between 0.1% and 2%.*

[Figure]

*Figure S 4 (A) Frequency of different classes of events in Payerne. (B) Boxplots of the growth rates of particles in the size ranges sub-3, 3- 7 nm and 7 – 15 nm, calculated using the 50% appearance time method from the NAIS positive ions. The pink diamonds are the mean value of the distribution. The red line represents the median of the data included in each box and the lower and upper edges of the box represent 25th and 75th percentiles of the data, respectively. The length of the whiskers represents 1.5× interquartile range which includes 99.3 % of the data. Data outside the whiskers are considered outliers and are marked with red crosses.*

**Reviewer # 3**

The manuscript of L. Dada and coauthors sheds light on the contribution, seasonality and diurnal behavior of the sources of ultrafine particles in a rural area in Switzerland. They use novel methods to estimate the contribution of the primary sources to the total UFP and discuss adequately their limitations. They also provide a detailed analysis of the observed NPF events and their characteristics, and compare them to other similar studies, giving insights into the potential driving NPF mechanisms in the area. However, confirming these mechanisms would require incorporating future VOC measurements. Therefore, I recommend the manuscript for publication after the authors address the minor comments outlined below.

**Specific comments:**

1. Lines 42-44: Since you discuss the source apportionment studies focusing on PM composition, I believe it is worth mentioning that there are also valuable studies focusing on UFP source apportionment methods, which utilize the different patterns and shapes of the measured UFP size distributions. Some examples include the works of Rivas et al., 2020; Garcia-Marlès et al., 2024; Kalkavouras et al., 2024; Vörösmarty et al., 2024.

We modified the sentence as follows:

*Composition-based source apportionment studies are typically done on mass basis for e.g. $PM_{2.5}$ and $PM_{10}$ (particulate matter with diameters less than 2.5 and 10 μm, respectively), yet some studies focused on UFP sources by number, utilizing the different patterns and shapes of the measured UFP size distributions (Grange et al., 2021; Trechera et al., 2023; Cai et al., 2024; Chen et al., 2022; Garcia-Marlès et al., 2024a; Garcia-Marlès et al., 2024b; Vörösmarty et al., 2024; Rivas et al., 2020; Kalkavouras et al., 2024).*

2. Section 3.3: It is not very clear how the event classification is being performed. Although the new tailored method (Fig. S3) is detailed and accounts for days difficult to interpret, it does not account for regional event days (since only local events are included). Do you first use the other methods mentioned (Dada et al., 2018; Dal Maso et al., 2005) for the initial classification, mainly of the regional events (banana plots), and then apply the new method for a more detailed analysis of the rest of the days?

We apologize for this misunderstanding. Indeed, we base our classification on the original method by Dal Maso et al. yet include examining the size distribution in the sub-3 nm range. Regional events could be both (1) local + transported, i.e. we observe the *banana* starting from the sub-3 nm diameters and growing into the shape of the banana where it merges with regional NPF events, or (2) only transported to our site where particles are observed as bananas that do not necessarily have a tail extending to the sub 3 nm region. In the case of Payerne, we did not observe any bananas that did not extend to the sub 3 nm size range. We clarify this in the text as follows:

Line 192: *Days in Payerne were classified into NPF event or nonevent days depending on the evolution of their particle number size distributions. Here, we followed the traditional method introduced by Dal Maso et al. (2005), and modified by Dada et al. (2018) to distinguish between local and transported events based on the size distributions below 3 nm. Here, we combined both aforementioned methods yet tailored them to fit our measurement location*

*better, which is subject to traffic emissions, as the previous two methods were developed for the boreal forest environment.*

*...*

Line 207: *We note that, the traditional regional events known as 'banana-shaped' events could be either (a) local + transported, i.e. we observe the evolution of the particles starting from the sub-3 nm diameters and growing into the shape of the banana where they merge with regional NPF events, or (b) only transported to our site where they are observed as a regional event but do not have a tail extending to the sub 3 nm region. In the case of Payerne, we did not observe any events that did not extend to the sub 3 nm size range, and hence all regional events are a combination of local nucleation and transported events.*

In section: 4.4.1 we also mention the following:

*We note that we did not observe any transported events at the site, without a local NPF event.*

3.  Lines 312-321: The ammonia's diurnal profile (Fig. 1D) during spring is very interesting. I guess the different behavior compared to the other seasons (earlier morning peak, as well as an increase during nighttime) is related to the fertilization process. It would be nice to comment on this a bit more. Why is the peak observed earlier compared to the other seasons? Why does the ammonia concentration increase during nighttime in spring and not during the other seasons? Does it have to do with the boundary layer development?

We modified the text to include the information on the distinct diurnal $NH_3$ cycle in spring as follows:

*In Payerne, the highest $NH_3$ concentrations are observed in spring, with a sharp peak in March when farmers prepare the first fertilization after winter and snow melt (Fig. 1, Fig. S7). In addition, warmer temperatures and longer days promote the release of ammonia (Fig. S10) (Pedersen et al., 2021). The distinct diurnal pattern of $NH_3$ concentrations during spring, i.e. earlier morning peak and increase during nighttime, could be attributed to farming activities such as fertilization and grazing, which peak at dawn and dusk.*

4.  Section 4.2: What was the concentration of $PM_{2.5}$ (and/or $PM_1$) on average during the campaign? It is important to have an idea about the mass concentration when studying UFP because they can affect the particle number (for example by increasing the CS).

Hourly $PM_{2.5}$ ranged between 0.10 and 33.6 µg/m$^3$ with an average of 9.2 ± 8.0 µg/m$^3$. We added this information to section 2.1.

5.  Section 4.3.1: The authors address adequately the limitations of the method. However, some studies have found that particles from NPF may contain non-volatile material in the examined temperature (300-350 °C). For example, Wu et al. (2017) reported that although a significant mass had evaporated after the NPF particles were exposed to a temperature of 300 °C, they still had a non-volatile core that was measured by their instrument. I believe the authors should also acknowledge that this method may potentially confuse the non-volatile fraction of the secondary material with primary material (currently only the opposite is mentioned which is also valid).

Thank you for pointing this out. However, the catalytic stripper method is much more efficient in removing volatile compounds than the TD method that was e.g. used in the Wu et al. study mentioned above. We added the following as part of the method limitation (Ln. 413):

*However, with a smaller likelihood and under certain circumstances, the uncertainty of the primary fraction could be further compounded the opposite way by the presence of very low volatile secondary particles that survive temperatures greater than 350°C (Kalberer et al., 2004; Wu et al., 2017).*

6. Lines 450-452: This is interesting and implies that there are higher traffic emissions during spring and summer compared to the winter and autumn. However, this behavior is not observed in the $NO_2$ diurnal profiles (Fig. 1B) where summer and spring have smaller concentrations compared to winter and autumn. Could you elaborate?

The seasonality of primary particles is different than that of $NO_2$, where the particles show higher concentrations in summer and spring compared to $NO_2$ which shows lowest concentrations in summer. Previous measurements in Payerne and Taenikon, another rural location in Switzerland, show lowest $NO_2$ concentrations in summer and attributed this observation to higher temperatures, increased photochemistry, lower wind speed and reduced emission sources in summer compared to the rest of the seasons (Steinbacher et al., 2007). Overall, the observed $NO_2$ diurnal cycles (and their seasonality) are a result of a complex interplay of emissions, meteorology, like dilution through advection and convection, (photo-) chemical processes during day and night, as well as removal processes such as the wet and dry deposition of $HNO_3$. As for the particles, it appears that they are more affected by traffic counts, which are much higher in summer compared to winter (Fig. R4), and less affected by the same losses as $NO_2$.

The text has been modified for $NO_2$ emission sources and losses in section 4.1.2 as per suggestion from both reviewers 1 and 3. Please refer to reviewer 1 comments #15. We also modified the text related to the non-volatile particles as per suggestion from reviewer 3 as follows:

*The primary nucleation mode particles are about two times higher in concentration in spring and summer compared to the colder seasons, with two peaks related to traffic rush hours as observed for the UFPs (Rönkkö et al., 2017), consistent with increased traffic volume in summer (FEDRO, 2021).*

[Figure]

*Figure R 5 Traffic volume at PAYERNE N (AR) highway A 1 closest to our measurement station (FEDRO, 2021).*

7. Lines 497-499: Does the term "local events" include both regional events and events taking place on a smaller spatial scale according to your classification? I understand that by "local"

you refer to NPF occurring on-site, with small ions appearing and growing to larger sizes. However, the word "local" is mainly used to describe NPF events taking place in a rather small spatial scale (Kerminen et al., 2018).

We clarified the distinction between local, transported, and regional events in section 3.3. Please refer to the response related to section 3.3 above.

8. Lines 537-539: I agree with this observation and it's interesting to see that ammonia (or amines) in this area is not the limiting factor. Could you elaborate on the statistical testing methods you used to confirm that sulfuric acid levels were significantly higher on NPF days compared to non-event days?

Since the distribution of the sulfuric acid concentration is not normally distributed (Fig. 6C), we opted to test significance using both you applied both the Wilcoxon Rank-Sum Test (considering completely independent samples) and the Wilcoxon Signed-Rank test (considering paired samples as the measurements on event and non-event days were at the same location). Both tests resulted in a $p < 0.05$ which shows that the sulfuric acid concentrations are significantly different between NPF and non-event days in Payerne during our measurement period.

The information is now integrated in the text:

*At Payerne, NPF events are observed on days with significantly enhanced of $H_2SO_4$ concentrations (calculated as a proxy from $SO_2$, see Sect. 3.4 - median events = 1.88 x $10^6$ cm$^{-3}$, median nonevents 1.07 x $10^6$ cm$^{-3}$, $p < 0.05$ using Wilcoxon's Rank-Sum and Signed-Rank tests) while $SO_2$ levels are similar on event and nonevent days (Fig. 6C, Fig. S22).*

9. Line 626: In line 421 you mention that about 25% of the UFP during the winter originates from primary sources (I assume that this fraction is even smaller for the other seasons), leaving the rest 75% to be of secondary origin. However, you find (line 626) that the overall NPF contribution to the UFP was only 5.07%. How is the rest 60-70% of UFPs explained? Are they also of secondary origin but are transported from other areas (background)? What areas/nearby cities influence the site? Are they a result of chemical processing of background aerosol particles (both primary and secondary)? I believe that this remaining fraction of UFPs is significant and deserves some additional discussion.

We thank the reviewer for this comment. In the context of our study, primary refers to particles that have an extremely low volatility or are BC particles. In our case, these particles account for 25% of the total UFP in winter. The remaining 75% are referred to as secondary in this definition. The 'core' of these 75% secondary particles could be either primary or secondary. The secondary 'core' ~ 5% are those produced via NPF, while the remaining 70% are as mentioned by the reviewer particles of any origin (traffic, biomass burning, long range transport…etc) that were subject to chemical processing. Referring to Fig. 5D, there are secondary particles concurrent with traffic emissions (1 hour later than the primary ones). These are likely secondary particles with a primary core resulting from traffic emissions and subject to processing on their way to our measurement location.

Accordingly, we clarified this terminology in section 4.3.2 to avoid any misunderstanding.

In addition, we performed a direct comparison between the primary particle counts from the BC method and non-volatiles method and is now included in the manuscript. Please also refer to our answers to reviewer 2.

**Technical comments:**

10. Line 36: Is this the correct citation of the Seinfeld and Pandis book? Maybe you mean 2016 (3$^{rd}$ edition) or 2006 (2$^{nd}$ edition)?

Thank you. Corrected.

11. Line 101: Please replace "are" with "is".

Thank you. Modified.

12. Line 137: Was this factor stable with time? What was its variation?

The 3.5 factor was chosen based on the cumulative distribution function at 50% of the ratio of NAIS to SMPS mean number concentration in the overlapping size range between both instruments (Fig. R6). The 25$^{th}$ and 75$^{th}$ percentiles were 2.5 and 5.8, respectively. The factor did not vary as a function of time nor particle concentration, therefore for consistency we opted for applying a constant factor to the whole data set. We added this information to the methods section.

[Figure]

*Figure R 6 A histogram showing the ratio of the dNdlogdp measured by the NAIS and SMPS in the overlapping size range between both instruments.A cumulative distribution function is shown on the right axis.*

13. Lines 178-180: The authors should consider replacing the terms $N_1$ and $N_2$ with alternatives such as $N_{primary}$ and $N_{secondary}$, respectively, to avoid confusion (both here and elsewhere), as these terms also represent the number of particles larger than 1 nm or 2 nm. Also, the total number of particles ($N$) refers to the total (combined NAIS corrected and SMPS) in the size range of 2.5 – 470 nm?

We modified the nomenclature of N primary and secondary in the equations as suggested by the reviewer. The N refers to the number concentration within the mode in question, therefore it could be $N_{ultrafine}$, $N_{aitken}$ or $N_{nucleation}$ depending on the size fraction we are considering. We clarified this in the methods section as well.

14. Fig. S10: Correct legend, the "x" is missing on the equation

Thanks. Modified.

15. Fig. S18A: The plot does not have axis labels and numbers.

We apologize for this problem which seems to be associated with uploading the files to the system. The original figure is as shown below. We will ensure a better quality of the supplementary material prior to the next upload.

[Figure]

*Figure S 5 (A) Frequency of different classes of events in Payerne. (B) Boxplots of the growth rates of particles in the size ranges sub-3, 3- 7 nm and 7 – 15 nm, calculated using the 50% appearance time method from the NAIS positive ions. The pink diamonds are the mean value of the distribution. The red line represents the median of the data included in each box and the lower and upper edges of the box represent 25th and 75th percentiles of the data, respectively. The length of the whiskers represents 1.5× interquartile range which includes 99.3 % of the data. Data outside the whiskers are considered outliers and are marked with red crosses.*

16. Lines 520-521: I guess the CS values reported are missing an " $\times 10^{-3}$ ".

Thank you. Modified.